# Emergent universal long-range structure in random-organizing systems

Satyam Anand [1,2,5] ✉, Guanming Zhang [2,3,5] ✉ & Stefano Martiniani [1,2,3,4] ✉

Self-organization through noisy interactions is ubiquitous across physics, mathematics, and machine learning, yet how long-range structure emerges from local noisy dynamics remains poorly understood. Here, we investigate three paradigmatic random-organizing particle systems drawn from distinct domains: models from soft matter physics (random organization, biased random organization) and machine learning (stochastic gradient descent), each characterized by distinct sources of noise. We discover universal long-range behavior across all systems, namely the suppression of long-range density fluctuations, governed solely by the noise correlation between particles. Furthermore, we establish a connection between the emergence of long-range structure and the tendency of stochastic gradient descent to favor flat regions of energy landscape—a phenomenon widely observed in machine learning. To rationalize these findings, we develop a fluctuating hydrodynamic theory that quantitatively captures all observations. Our study resolves long-standing questions about the microscopic origin of noise-induced hyperuniformity, uncovers striking parallels between stochastic gradient descent dynamics on particle system energy landscapes and neural network loss landscapes, and should have wide-ranging applications—from the self-assembly of hyperuniform materials to ecological population dynamics and the design of generalizable learning algorithms.

While typically associated with disorder, noise can paradoxically drive the emergence of diverse forms of order, such as pattern formation[1], self-organization[2,3], suppression of chaos[4], selection of ordered states[5], and swarming[6]. Physical systems exhibit a broad spectrum of order: perfect crystals and ideal gases mark the extremes, while intermediate regimes display correlated disorder, such as hyperuniformity—where local disorder coexists with the anomalous suppression of long-range density fluctuations[7]. Hyperuniformity can emerge either at criticality[8–10], or away from it[11–14]. Away from criticality, in equilibrium, hyperuniformity requires long-range interactions[7], whereas out of equilibrium, it can emerge from long- or short-range, and even noisy interactions[11–16]. The process by which long-range spatial structure

develops away from criticality—particularly in systems interacting solely via short-range, noisy dynamics—is a long-standing question that remains poorly understood.

Non-equilibrium particle systems with short-range noisy interactions—such as random organization (RO)[2,9,17–20], biased random organization (BRO)[10,12,14,19,21,22], and stochastic gradient descent (SGD)[23]—provide an ideal framework to investigate the noise-driven emergence of long-range spatial structure. These systems have been studied in a wide variety of contexts, such as sheared colloidal suspensions[2,9,24,25], random close packing[10,21], two-dimensional crystallization[14], and self-supervised learning[23,26]. All systems undergo a phase transition as the particle volume fraction increases; from a low-density state where all

[1]Courant Institute of Mathematical Sciences, New York University, New York, NY 10003, USA. [2]Center for Soft Matter Research, Department of Physics, New York University, New York, NY 10003, USA. [3]Simons Center for Computational Physical Chemistry, Department of Chemistry, New York University, New York, NY 10003, USA. [4]Center for Neural Science, New York University, New York, NY 10003, USA. [5]These authors contributed equally: Satyam Anand, Guanming Zhang. ✉e-mail: sa7483@nyu.edu; gz2241@nyu.edu; sm7683@nyu.edu

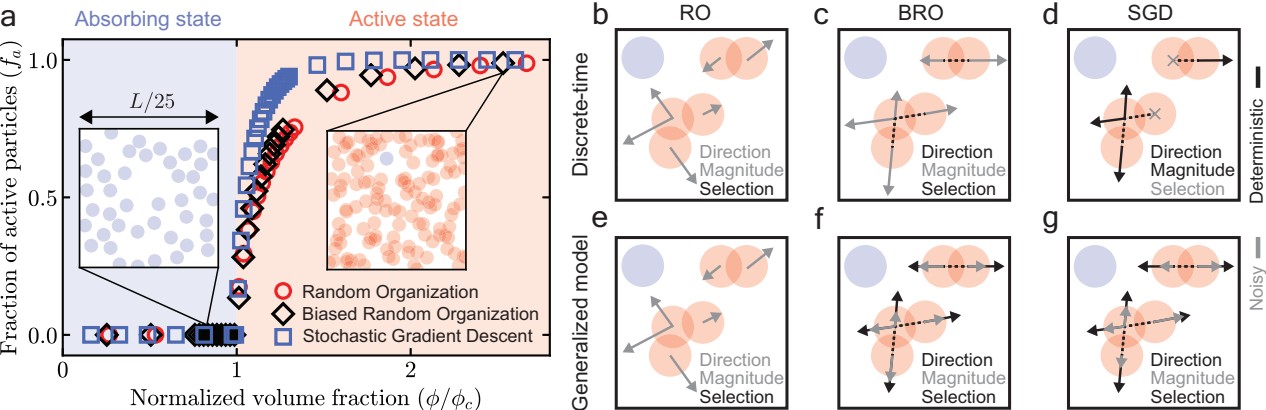

**Fig. 1 | Random-organizing systems. a** Absorbing-to-active phase transition. The steady-state fraction of active (overlapping) particles ($f_a$) plotted as a function of normalized volume fraction $\phi/\phi_c$, where $\phi_c$ is the critical volume fraction. Starting from an initial random configuration, at time $t \to \infty$, in the absorbing state ($\phi/\phi_c < 1$), $f_a = 0$, while in the active state ($\phi/\phi_c > 1$), $f_a > 0$. Insets show zoomed in exemplar configurations of discrete-time BRO, and $L$ is the side length of the simulation box. Blue particles have no overlapping neighbors whereas red particles have at least one overlapping neighbor. Schematics of discrete-time RO (**b**) (Eq. (1)), BRO (**c**)

(Eq. (2)), SGD (**d**) (Eq. (3)), and the corresponding continuous-time approximation (generalized model, Eq. (5)) of RO (**e**), BRO (**f**), and SGD (**g**). Solid gray colors denote noise, while solid black colors denote deterministic interactions. Dashed black lines connect the centers of a pair of overlapping particles. Crosses in panel (**d**) denote unselected particles. For RO, the direction and the magnitude of the kicks are both noisy; for BRO, only the magnitude of the kicks are noisy; while for SGD, only the selection of active particles is noisy. Notice that in the generalized model (Eq. (5)), the selection noise of SGD is approximated by a noise in the magnitude of the kicks.

motion ceases (absorbing state), to a high-density state where motion persists forever (active state)[27–29]. Irrespective of microscopic details, RO, BRO, and SGD belong to the same universality class, i.e., display the same critical behavior[10,20,23]. Away from criticality in the active phase, however, variations in microscopic interactions significantly influence the emergent long-range structure[12,14,17,18]. Despite extensive experimental, numerical, and theoretical research over two decades, a quantitative microscopic understanding of dynamics and structure far from criticality remains elusive[2,9,10,12,14,17–19,21–24]. Fundamental questions remain unanswered: How does macroscopic structure emerge from noisy interactions? Moreover, what universal principles govern the variability in emergent structures within and across different random-organizing systems? Finally, is the emergent long-range structure in SGD related to its ability to discover flat regions of energy landscape—a feature linked to robust generalization in machine learning[30]?

Here, combining particle and continuum simulations with hydrodynamic theory, we provide a quantitative, microscopic understanding of the active phases of RO, BRO, and SGD—random-organizing systems with distinct sources of noise. We discover universal long-range behavior across all three systems governed by a single parameter: the noise correlation coefficient between particles. All systems self-organize to suppress density fluctuations below a crossover length scale, which diverges as the noise becomes anti-correlated (reciprocal interactions), resulting in strong (class I[7]) hyperuniformity. Further, by directly coarse-graining the microscopic dynamics, we develop a fluctuating hydrodynamic theory for random-organizing systems that quantitatively predicts both the emergence of long-range structure and the crossover length scale across all systems. Finally, we demonstrate how noise correlation, batch size, and learning rate bias SGD towards flatter regions of energy landscape—a finding that aligns with empirical observations in machine learning[30–33]—and establish a connection between the emergence of long-range structure and the ability of SGD to generalize effectively. Our study underscores the critical role of noise correlations in facilitating long-range structure, and has wideranging applications—from providing a robust method for self-assembling hyperuniform structures, to offering insights into other systems having correlated noise, such as neural population activity in the brain[34], ecological population dynamics[35], and gene expression dynamics in cells[36].

## Results
### Setup
Random-organizing systems—RO, BRO, and SGD—are discrete-time systems consisting of $N$ interacting spherical particles of radius $R$ in $d$-dimensional space. At any given time-step, the positions of isolated particles (blue in Fig. 1b–d) do not evolve, and those of overlapping (active) particles (red in Fig. 1b–d) evolve according to system specific rules designed to resolve particle overlaps. All systems undergo an absorbing-to-active phase transition as the particle volume fraction $\phi$ increases (Fig. 1a). Starting from a random initial configuration, for $\phi < \phi_c$, the system finds an absorbing configuration ($f_a = 0$), whereas for $\phi > \phi_c$, the system never finds such a configuration and reaches a non-equilibrium steady state ($f_a > 0$) (Fig. 1a). Here, $f_a$ is the fraction of active particles in the system. We report all our results in the active phase when the system has reached steady state (Methods).

### Random organization
RO was originally introduced to model experiments on sheared colloidal suspensions at high Péclet number[2,24]. Consider a system undergoing periodic shear cycles. After every shear cycle, overlapping particles are given a "kick" in a random direction, and non-overlapping particles return to their original position[2,9]. RO was subsequently simplified to a model without external shearing, which retains all the essential properties of the original version[17–19,29]. Here, we study this simpler isotropic version of RO[17–19,29].

In RO, the dynamics of the position of particle $i$ at time-step $m+1$ ($\mathbf{x}_i^{m+1}$) is given by,

$$\mathbf{x}_i^{m+1} = \mathbf{x}_i^m + \epsilon \sum_{j \in \Gamma_i^m} u_{ji}^m \zeta_{ji}^m, \tag{1}$$

where $\epsilon$ controls the magnitude of the pairwise kick given by particle $j$ to $i$, $u_{ji}^m$ is a random number sampled from a standard uniform distribution ($U[0, 1]$) at time-step $m$, $\zeta_{ji}^m$ is a random unit vector sampled uniformly on the surface of a $d$-dimensional unit hypersphere at time-step $m$, and $\Gamma_i^m = \{j \mid |\mathbf{x}_j^m - \mathbf{x}_i^m| < 2R, j \neq i\}$ is the set containing all particles that overlap with particle $i$ at time-step $m$ (Fig. 1b). We set $\zeta_{ji}^m = -\zeta_{ij}^m$ so that particles effectively move away from ("repel") each other. Finally, $c \in [-1, 0]$ is the Pearson correlation coefficient between corresponding components of the complete noise vectors

$\omega_{ij,\alpha}^m = \epsilon u_{ij}^m \zeta_{ij,\alpha}^m$ and $\omega_{ji,\alpha}^m$, defined by $\langle \omega_{ij,\alpha}^m \omega_{ji,\beta}^m \rangle / \langle \omega_{ij,\alpha}^m \rangle \langle \omega_{ji,\alpha}^m \rangle = c\,\delta_{\alpha\beta}$. $c = 0$ denotes pairwise kicks which are uncorrelated, and $c = -1$ denotes pairwise kicks which are anti-correlated (equal in magnitude, $u_{ij}^m = u_{ji}^m$, and opposite in direction, i.e., conserve pairwise center of mass).

## Biased random organization

BRO was originally introduced to study random close packing of spheres[10,21]. Similar to RO, overlapping particles are given a kick, however, the kick is "biased" along the direction of the line joining the centers of overlapping particles (Fig. 1c). An interesting feature of BRO is that for $\epsilon \to 0$, its critical point was proposed as an alternative definition of random close packing[10,21], which, despite decades of research, still lacks a clear definition[37–41].

In BRO, the dynamics of the position of particle $i$ at time-step $m+1$ ($\mathbf{x}_i^{m+1}$) is given by,

$$\mathbf{x}_i^{m+1} = \mathbf{x}_i^m + \epsilon \sum_{j \in \Gamma_i^m} u_{ji}^m \hat{\mathbf{x}}_{ji}^m, \tag{2}$$

where $\epsilon$ and $u_{ji}^m$ are defined as in RO, and $\hat{\mathbf{x}}_{ji}^m = -(\mathbf{x}_j^m - \mathbf{x}_i^m)/|\mathbf{x}_j^m - \mathbf{x}_i^m|$ is the deterministic unit vector pointing from the center of particle $j$ to $i$ at time-step $m$ (Fig. 1c). $c \in [-1, 0]$ is the Pearson correlation coefficient between corresponding components of the complete noise vectors $\omega_{ij,\alpha}^m = \epsilon u_{ij}^m \hat{x}_{ij,\alpha}^m$ and $\omega_{ji,\alpha}^m$. Notice that $c = -1$ corresponds to $u_{ij}^m = u_{ji}^m$, since $\hat{\mathbf{x}}_{ji}^m = -\hat{\mathbf{x}}_{ij}^m$.

## Stochastic gradient descent

SGD is a widely used optimization algorithm, e.g., in artificial neural networks, to minimize a loss function composed of a sum of many terms[42]. Stochasticity in SGD comes from the random selection of a subset of terms in the sum at every step. In the context of interacting particle systems, we take the loss to be the total energy $E = \sum_i \sum_{j \geq i} V(\mathbf{x}_i, \mathbf{x}_j)$, where $V(\mathbf{x}_i, \mathbf{x}_j)$ is any pairwise potential. SGD then corresponds to randomly selecting a subset of terms in $E$ and updating the corresponding particle positions–either one or both at once–to minimize the partial energy. This is in contrast to simultaneously moving all active particles, which corresponds to (noiseless) gradient descent[23]. Notice that our approach of SGD applied on particle systems is unlike previous works relating neural networks to particle systems by treating parameters (weights) as interacting particles[43].

In SGD, the dynamics of the position of particle $i$ at time-step $m+1$ ($\mathbf{x}_i^{m+1}$) is given by,

$$\mathbf{x}_i^{m+1} = \mathbf{x}_i^m - \alpha \sum_{j \in \Gamma_i^m} \theta_{ji}^m \nabla_i V_{ji}^m, \tag{3}$$

where $\nabla_i = \nabla_{\mathbf{x}_i}$, $\alpha$ is the learning rate having units of length/force, $V_{ji}^m = V(|\mathbf{x}_j^m - \mathbf{x}_i^m|)$ is the pairwise interaction potential between particles $i$ and $j$, and $\theta_{ji}^m$ is a random number sampled from a Bernoulli distribution having parameter $b_f$ (batch fraction) at time-step $m$ (Fig. 1d). $b_f$ represents the average fraction of active particle pairs $(i, j)$ that move at any given time-step. $c \in [-1, 0]$ is the Pearson correlation coefficient between corresponding components of the complete noise vectors $\omega_{ij,\alpha}^m = -\alpha \theta_{ij}^m \partial_{i,\alpha} V_{ij}^m$ and $\omega_{ji,\alpha}^m$ originating from the pairwise correlated selection noise $\theta_{ij}^m$. Notice that since $\nabla_i V_{ji}^m = -\nabla_j V_{ji}^m$, $c = -1$ corresponds to $\theta_{ij}^m = \theta_{ji}^m$. While $V_{ji}$ can be any short- or long-range potential in SGD, here, we consider a class of short-range, repulsive potentials given by

$$V_{ij}(r) = \begin{cases} \frac{\mathcal{E}}{p}\left(1 - \frac{r_{ij}}{2R}\right)^p, & \text{if } 0 < r_{ij} < 2R, \\ 0, & \text{otherwise,} \end{cases} \tag{4}$$

where $r_{ij} = |\mathbf{x}_j - \mathbf{x}_i|$, $\mathcal{E}$ is the characteristic energy scale, and $p$ controls the stiffness of the potential.

## Universal active phase behavior

There are three distinct sources of noise in random-organizing systems: magnitude of kicks, direction of kicks, and selection of particles. Notice that the origin of stochasticity in RO, BRO, and SGD is different; (i) for RO, the magnitude and direction of kicks are both noisy, while the selection of particles is deterministic, (ii) for BRO, the magnitude of kicks is noisy, while both the direction of kicks and selection of particles is deterministic, (iii) for SGD, the selection of particles is noisy, while the magnitude and direction of kicks are both deterministic (Fig. 1b–d, Eqns. (1), (2), and (3)). We perform particle simulations for RO, BRO, and SGD in the active phase ($\phi > \phi_c$) and measure the long-range structure, quantified by the radially averaged structure factor $S(k)$ and variance in number density $\delta\rho^2(l)$ (Methods). $l$ is the diameter of the hypersphere used for measuring density fluctuations and $k = 2\pi/l$ is the wave number. We study all properties above a threshold length scale $l_0 = 2\pi/k_0$, below which we find system-specific short-range behavior (Fig. 2b, c). Hereafter, we work with normalized quantities: $\tilde{l}$, $\tilde{k}$, $\tilde{S}(\tilde{k})$, and $\widetilde{\delta\rho}(\tilde{l})$ (see Fig. 2 caption for definitions).

Despite having microscopically different dynamics, all random-organizing systems display a universal long-range structure, controlled solely by the pairwise noise correlation $c$, and qualitatively independent of all other parameters–be it the kick magnitude $\epsilon$, volume fraction $\phi$, or spatial dimension $d$ in RO and BRO, or $\phi$, $d$, learning rate $\alpha$, batch fraction $b_f$, potential stiffness $p$, and energy scale $\mathcal{E}$ in SGD. (Fig. 2a, b, c, Supplementary Information (SI) Figs. S2, S3, S4). Remarkably, the active phase behavior across all systems is independent of $d$, in contrast to their critical behavior, which is heavily dependent on $d$ [9,10,23] (Figs. S2c, S3c, S4c). All systems self-organize to suppress density fluctuations below a normalized crossover length scale $\tilde{l}_c$; specifically, for length scales $1 < \tilde{l} < \tilde{l}_c$, the structure factor follows a power law ($\tilde{S} \sim \tilde{k}^2$ and $\widetilde{\delta\rho}^2 \sim \tilde{l}^{-(d+1)}$), whereas for $\tilde{l} > \tilde{l}_c$, the structure factor is constant, ($\tilde{S} \sim \text{const.}$) and $\widetilde{\delta\rho}^2 \sim \tilde{l}^{-d}$ (Fig. 2b, c). Further, the crossover length scale $\tilde{l}_c$ increases monotonically as $c$ decreases: as the pairwise noise becomes more negatively correlated, density fluctuations are suppressed up to larger length scales (Fig. 2b inset). Consequently, the infinite wavelength density fluctuations, $\widetilde{\delta\rho}^2(\tilde{l} \to \infty) \propto \tilde{S}(\tilde{k} \to 0)$, decrease monotonically as $c$ decreases (Fig. 2c inset). Finally, when the noise is anti-correlated ($c = -1$), the crossover length scale $\tilde{l}_c \to \infty$ and the system becomes strongly hyperuniform, $\tilde{S}(\tilde{k} \to 0) \sim \tilde{k}^2$, and $\widetilde{\delta\rho}^2(\tilde{l} \to \infty) \sim \tilde{l}^{-(d+1)}$ (Fig. 2b, c).

Our results are consistent with previous studies on RO, which focused on the specific case of uncorrelated noise ($c = 0$)[17,18], and on BRO, which focused on the specific case of anti-correlated noise ($c = -1$)[12,14]. So, why do microscopically distinct systems–RO, BRO, and SGD–exhibit universal long-range behavior?

## Generalized model

We now develop a continuous-time model of discrete-time random-organizing systems. Using the framework of stochastic modified equations[23,44], we approximate the discrete-time dynamics by a continuous-time stochastic differential equation (SDE) (SI Sec. I.A). In the resulting genralized model, the dynamics of the position of particle $i$ ($\mathbf{x}_i$) is given by an overdamped Langevin equation,

$$\frac{d\mathbf{x}_i(t)}{dt} = -\frac{1}{\gamma} \sum_{j=1}^N \nabla_i V_{ji} + \sum_{j=1}^N \sqrt{\mathbf{\Lambda}_{ji}} \cdot \boldsymbol{\xi}_{ji}, \tag{5}$$

where $\gamma$ is the friction constant, $V_{ji}$, $\mathbf{\Lambda}_{ji}$ are short-range, pairwise interaction potential and diffusion matrix between particles $j$ and $i$,

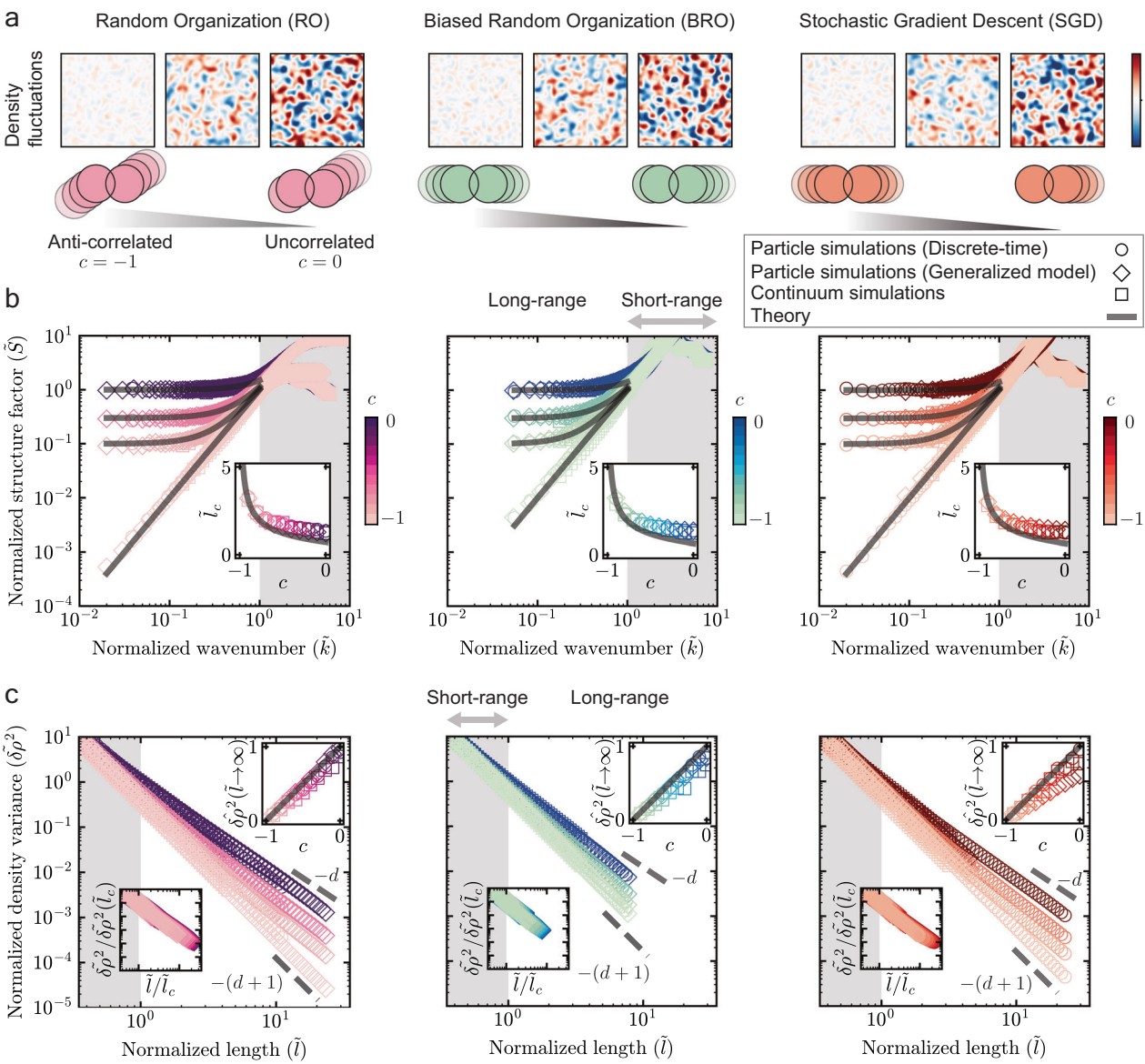

**Fig. 2 | Universal long-range structure in random-organizing systems. a** Coarse-grained density fluctuations $\delta\rho(c)/|\delta\rho_{\text{avg}}(c = 0)|$ in random-organizing systems, where $c$ is the pairwise noise correlation, and $\delta\rho_{\text{avg}}$ denotes average density fluctuations over the whole system. The three panels denote $c = -1, c = -0.75, c = 0$ (left to right) for each system. A Gaussian kernel of width $100R$ was chosen for coarse-graining all systems, where $R$ is the particle radius. As the pairwise noise correlation $c$ goes from 0 (uncorrelated) to $-1$ (anti-correlated), the density fluctuations are suppressed for all systems. **b** Normalized radially averaged structure factor $\widetilde{S}(\tilde{k})$ versus normalized wave number $\tilde{k}$ for RO, BRO, and SGD (left to right). $\widetilde{S} = S(k)/S_0(2\pi/L)$ where $S_0(2\pi/L)$ is the structure factor for $c = 0$ at $k = 2\pi/L$, and $L$ is the side length of the simulation box. $\tilde{k} = k/k_0$, where $k_0$ is the value at which $\widetilde{S}(k_0) = 1$ for anti-correlated noise ($c = -1$) of the same system. Solid black lines show predictions of Eq. (10) for different values of $c$. Inset shows the normalized

crossover length scale ($l_c/l_0 = \tilde{l}_c = 1/\tilde{k}_c$) versus $c$. We find the normalized crossover wavenumber ($\tilde{k}_c$) in simulations as the intersection, on a log-log plot, between a fit of slope 0 near the $\tilde{k} \to 0$ region, and a fit of slope 2 near the $\tilde{k} \approx 1$ region. Solid black line in the inset shows prediction of Eq. (11). Gray shaded regions denote short-range behavior ($\tilde{k} > 1$). **c** Normalized variance of number density $\widetilde{\delta\rho}^2(\tilde{l})$ versus normalized diameter of the hypersphere ($\tilde{l}$) used for measuring density fluctuations for RO, BRO, and SGD (left to right). $\widetilde{\delta\rho}^2(\tilde{l}) = \delta\rho^2(l)/\delta\rho^2(l_0)$ where $\delta\rho^2(l_0)$ is the density variance for $c = 0$ at $l = l_0$, and $\tilde{l} = l/l_0$, where $l_0 = 2\pi/k_0$. Bottom inset shows data collapse of density variances when $\tilde{l}$ is rescaled by $\tilde{l}_c$, and $\widetilde{\delta\rho}^2(\tilde{l})$ is rescaled by $\widetilde{\delta\rho}^2(\tilde{l}_c)$. Top inset shows infinite wavelength density fluctuations $\widehat{\delta\rho}^2(\tilde{l} \to \infty)$ versus $c$. $\widehat{\delta\rho}^2(c) = [\widetilde{\delta\rho}^2(c) - \widetilde{\delta\rho}^2(c = -1)]/\widetilde{\delta\rho}^2(c = 0)$. Solid black line denotes $1 + c$ (prediction of Eq. (10) in the limit $\tilde{k} \to 0$). Gray shaded regions denote short-range behavior ($\tilde{l} < 1$).

respectively, $\sqrt{\Lambda_{ji}}$ denotes a matrix square root, and Eq. (5) is interpreted in the Itô sense. $\boldsymbol{\xi}_{ji}$ is a pairwise, Gaussian noise given by the particle $j$ to particle $i$ having mean $\langle\xi_{ji,\alpha}(t)\rangle = 0$ and covariance matrix $\langle\xi_{ij,\alpha}(t)\xi_{kl,\beta}(t')\rangle = \delta(t - t')\,\delta_{\alpha\beta}(\delta_{ik}\delta_{jl} + c\,\delta_{il}\delta_{jk})$, where $c \in [-1, 0]$ is the Pearson correlation coefficient between $\xi_{ij,\alpha}(t)$ and $\xi_{ji,\alpha}(t)$. Eq. (5), supplemented with system-specific $\gamma$, $V_{ij}$ and $\Lambda_{ij}$, is an SDE approximating RO, BRO, and SGD (SI Sec. I.A, Table S1). Notice that in the generalized

model, the source(s) of noise for RO and BRO remain the same, while the selection noise in SGD becomes a noise on the magnitude of the kicks (Fig. 1e–g, SI Sec. I.A)[23].

We perform particle simulations of the generalized model for RO, BRO, and SGD and find that the long-range structure is quantitatively the same as that for their discrete-time counterparts (Methods, Fig. 2b, c). Thus, the generalized model serves as an accurate continuous-time approximation for all random-organizing systems.

## Fluctuating hydrodynamic theory

Equipped with the generalized model, we formulate a theory for the evolution of the density field $\rho(\mathbf{x}, t)$. Dean's method, originally introduced for Brownian particles with additive noise[45], and later extended to study systems with multiplicative noise[46–48], is a well-known approach for directly coarse-graining microscopic dynamics. However, it has not yet been extended to systems where noise is both pairwise and correlated across components. Starting from Eq. (5), we extend Dean's method[45] and its subsequent generalizations[48] to incorporate pairwise correlated noise, and derive the resulting fluctuating hydrodynamic equation for $\rho(\mathbf{x}, t)$ in arbitrary spatial dimension $d$ to get (SI Sec. I.B.2),

$$\frac{\partial \rho(\mathbf{x}, t)}{\partial t} = \underbrace{- \nabla \cdot [\rho(\mathbf{x})\mathbf{v}(\mathbf{x})]}_{\text{drift term}} + \underbrace{\nabla\nabla : [\mathbf{D}(\mathbf{x})\rho(\mathbf{x})]}_{\text{diffusion term}} - \underbrace{\nabla \cdot \mathbf{j}_n(\mathbf{x})}_{\text{noise term}}, \quad (6)$$

where $\nabla = \nabla_{\mathbf{x}}$, and : denotes a double dot product.

The velocity $\mathbf{v}(\mathbf{x})$ in Eq. (6) is given by

$$\mathbf{v}(\mathbf{x}) = -\frac{1}{\gamma} \langle \nabla V(\mathbf{x}, \mathbf{y}) \rangle_{\rho(\mathbf{y})}, \quad (7)$$

where $\langle a \rangle_{\rho(\mathbf{y})} = \int a\rho(\mathbf{y})d\mathbf{y}$, and $V(\mathbf{x}, \mathbf{y})$ is the "continuous" version of $V_{ji}$, given by replacing $\mathbf{x}_i$ and $\mathbf{x}_j$ by $\mathbf{x}$ and $\mathbf{y}$ in Eq. (4). $\mathbf{v}(\mathbf{x})$ originates from the deterministic (first) term in Eq. (5), and can be understood as the average force at point $\mathbf{x}$ due to the local interaction potential, $-\langle \nabla V(\mathbf{x}, \mathbf{y}) \rangle_{\rho(\mathbf{y})}$, divided by the friction coefficient $\gamma$.

The diffusion tensor $\mathbf{D}(\mathbf{x})$ in Eq. (6) is given by

$$\mathbf{D}(\mathbf{x}) = \frac{1}{2} \langle \mathbf{\Lambda}(\mathbf{x}, \mathbf{y}) \rangle_{\rho(\mathbf{y})}. \quad (8)$$

$\mathbf{D}(\mathbf{x})$ originates from the noise (second) term in Eq. (5), and can be understood as the average diffusion tensor over the local density.

The stochastic flux $\mathbf{j}_n(\mathbf{x})$ in Eq. (6) is given by

$$\mathbf{j}_n(\mathbf{x}) = -\sqrt{\rho(\mathbf{x})} \int \sqrt{\rho(\mathbf{y})}\sqrt{\mathbf{\Lambda}(\mathbf{x}, \mathbf{y})} \cdot \boldsymbol{\eta}(\mathbf{x}, \mathbf{y}) \, d\mathbf{y}, \quad (9)$$

where $\sqrt{\mathbf{\Lambda}}$ denotes a matrix square root, and $\mathbf{\Lambda}(\mathbf{x}, \mathbf{y})$ is the "continuous" version of $\mathbf{\Lambda}_{ji}$. $\boldsymbol{\eta}(\mathbf{x}, \mathbf{y}, t)$ is a vectorial, two-point, Gaussian noise field having mean $\langle \eta_\alpha(\mathbf{x}, \mathbf{y}, t) \rangle = 0$, and covariance matrix $\langle \eta_\alpha(\mathbf{x}, \mathbf{y}, t)\eta_\beta(\mathbf{u}, \mathbf{w}, t') \rangle = \delta_{\alpha\beta}\delta(t - t')[\delta(\mathbf{x} - \mathbf{u})\delta(\mathbf{y} - \mathbf{w}) + c\,\delta(\mathbf{x} - \mathbf{w})\delta(\mathbf{y} - \mathbf{u})]$, where $c \in [-1, 0]$ is the Pearson correlation coefficient between $\eta_\alpha(\mathbf{x}, \mathbf{y}, t)$ and $\eta_\alpha(\mathbf{y}, \mathbf{x}, t)$. $\mathbf{j}_n(\mathbf{x})$ originates from the noise (second) term in Eq. (5). Notice that the noise at any spatial location $\mathbf{x}$ can be viewed as the sum of independent kicks from $n_y \propto \rho(\mathbf{y})\delta V$ particles at $\mathbf{y}$ given to $n_x \propto \rho(\mathbf{x})\delta V$ particles at $\mathbf{x}$, where $\delta V$ is an infinitesimal volume. Then, since the sum of $n$ Gaussian noises yields a Gaussian noise with standard deviation $\propto \sqrt{n} \propto \sqrt{\rho}$, we have $\mathbf{j}_n \propto \sqrt{\rho(\mathbf{x})}\sqrt{\rho(\mathbf{y})}$. Further, similar to Eq. (5), $\sqrt{\mathbf{\Lambda}(\mathbf{x}, \mathbf{y})}$ acts as a projection tensor, making the noise anisotropic. Finally, integrating over $\mathbf{y}$ collects contributions from different locations to the total stochastic flux at $\mathbf{x}$.

We perform finite-difference simulations of Eq. (6) for RO, BRO, and SGD, and find that the long-range structure is quantitatively the same as particle simulations (Methods, Fig. 2b, c). Thus, our coarse-grained theory quantitatively captures the long-range structure for all random-organizing systems.

To gain further analytical insights on Eq. (6), we linearize $\rho(\mathbf{x}, t)$ to first order and derive the analytical form of the static structure factor (SI Sec. I.B.2),

$$\widetilde{S}(\widetilde{k}) = (1 + c) + [M(1 + c) - c]\widetilde{k}^2, \quad (10)$$

where $M$ is a known system-dependent constant (Eqs. S64, S66, and S68, SI Sec. I.B.2). Eq. (10) directly relates $c$, the microscopic noise correlation coefficient between a pair of particles, to the long-range structure in the system, and quantitatively predicts $S(k)$ for all random-organizing systems without free parameters (Fig. 2b). Further, since $\widetilde{\delta\rho}^2 (l \to \infty) \propto \widetilde{S}(\widetilde{k} \to 0)$[7], Eq. (10) also predicts the behavior of infinite wavelength density fluctuations for all random-organizing systems (Fig. 2c inset).

Equation (10) shows a competition between two terms. The first term $(1 + c)$ makes the long-range structure random, while the second term $[M(1 + c) - c]\widetilde{k}^2$ makes the long-range structure strongly hyperuniform. The ratio of these two competing terms gives the normalized crossover length scale,

$$\widetilde{l}_c = \sqrt{M - \frac{c}{1 + c}}, \quad (11)$$

which quantitatively predicts the crossover length scale for all random-organizing systems (Fig. 2b inset).

Two remarks are in order. First, in a variety of other systems having local center of mass conserving dynamics[12,49–52] (equivalent to $c = -1$ in random-organizing systems) and displaying hyperuniformity in the dense phase, the coarse-grained noise term is of Laplacian form $\nabla^2[\sqrt{\rho(\mathbf{x})}\omega(\mathbf{x})]$, where $\omega(\mathbf{x})$ is a spatially uncorrelated, one-point Gaussian noise field. In contrast, the noise term in our microscopically derived theory (Eqs. (6), (9)) is fundamentally different from "Laplacian noise" in two ways: (i) it enters Eq. (6) in the "divergence" form and, (ii) is spatially correlated. Second, the widely used Fokker-Planck coarse-graining approach gives a density evolution equation identical to Eq. (6), but without the stochastic flux term ($\mathbf{j}_n(\mathbf{x}) = \mathbf{0}$), which, after density linearization, trivially predicts spatially uniform steady-state density ($\rho(\mathbf{x}) = $ constant and $S(k) = 0$) independent of $c$ (SI Sec. I.B.1). This demonstrates the crucial role of noise and noise correlations even at the coarse-grained scale in determining the long-range structure, thereby underscoring the importance of our theoretical framework.

## Flatness of energy landscape in stochastic gradient descent

SGD is widely used for training neural networks, not only for its computational efficiency but also for its remarkable ability to steer neural networks toward flat regions of their loss landscapes[32,42,53]. This attribute is crucial for "good" learning algorithms since flatter regions are strongly correlated with better generalization performance on unseen data[30–33]. The bias towards flat regions, due to the selection noise in SGD, highlights the vital role of noise in shaping learning dynamics in neural networks. This, then, raises two questions: (i) Does SGD-driven descent of energy landscapes in particle systems similarly bias the dynamics toward flat regions, akin to neural networks? If so, (ii) Can this bias be linked to the long-range structure observed in these particle systems?

We first examine how the flatness of the explored regions of the energy landscape varies with noise correlation $c$, batch fraction $b_f$, and learning rate $\alpha$ in SGD (Eq. (3)). We add a small noise $\mathbf{N}$ to the steady-state configurations $\mathbf{X}$ obtained in particle simulations to get a perturbed configuration $\mathbf{X} + \Delta\mathbf{X}$ and measure the change in the total energy of the system, $\Delta E = \langle E(\mathbf{X} + \Delta\mathbf{X}) - E(\mathbf{X}) \rangle_{\mathbf{N}} \propto \text{Tr}(\mathbf{H})$, where $E = \sum_i \sum_{j>i} V_{ij}$, and $\mathbf{H}$ is the Hessian matrix of the system (Eq. (4), Fig. 3a, Methods)[23]. $\Delta E$ is a measure of the flatness of the energy landscape: lower $\Delta E$ corresponds to flatter regions (Fig. 3a)[23,30]. We find that $\Delta E$ decreases as $c$ decreases, meaning that increase in long-range structure leads to flatter regions (Figs. 3b, 2b). We now fix $c = -1$ (anticorrelated noise), and find that $\Delta E$ increases with $b_f$[23] and decreases with $\alpha$, suggesting that lower batch fractions and higher learning rates lead to flatter regions of energy landscape—consistent with results on SGD dynamics in neural networks (Fig. 3c, d)[30–32]. Notice that $c = -1$ and

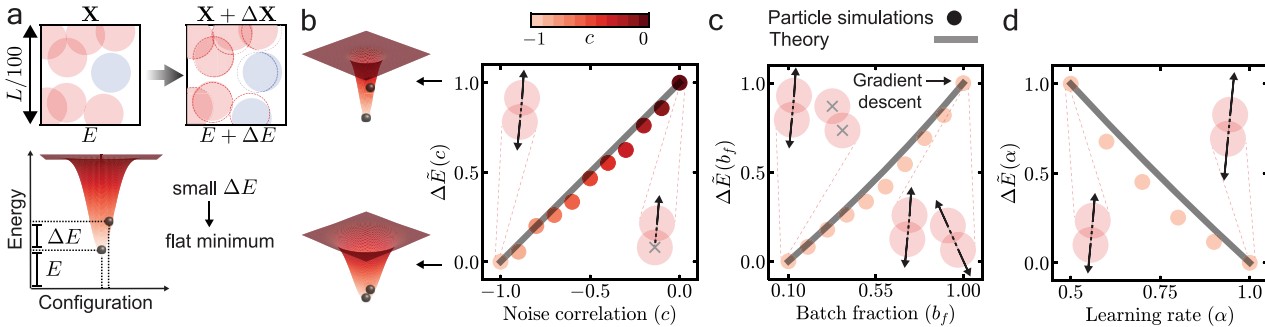

**Fig. 3 | Flatness of energy landscape in stochastic gradient descent (SGD).**
**a** Zoomed-in exemplar configuration of discrete-time SGD before and after adding a small noise to all particles (Top). Schematic of a system (black ball) in a steady-state configuration. The system has energy $E$ at the steady-state configuration $\mathbf{X}$, and energy $E + \Delta E$ at a slightly perturbed position $\mathbf{X} + \Delta \mathbf{X}$ after the addition of a small noise $\mathbf{N}$. **b** Normalized energy change $\Delta\tilde{E}(c)$ versus noise correlation $c$. $\Delta E(c)$ is normalized as: $\Delta\tilde{E}(c) = [\Delta E(c) - \Delta E(c = -1)]/[\Delta E(c = 0) - \Delta E(c = -1)]$. Black line shows prediction of Eq. S79 (SI Sec. I.C). Schematic showing two regions of energy landscape with different flatness (left). **c** Normalized energy change $\Delta\tilde{E}(b_f)$ versus batch fraction $b_f$. $\Delta E(b_f)$ is normalized as: $\Delta\tilde{E}(b_f) = [\Delta E(b_f) - \Delta E(b_f = 0.1)]/[\Delta E(b_f = 1.0) - \Delta E(b_f = 0.1)]$. Black line shows prediction of Eq. S82 (SI Sec. I.C). **d** Normalized energy change $\Delta\tilde{E}(\alpha)$ versus learning rate $\alpha$. $\Delta E(\alpha)$ is normalized as: $\Delta E(\alpha) = [\Delta E(\alpha) - \Delta E(\alpha = 1.0)]/[\Delta E(\alpha = 0.5) - \Delta E(\alpha = 1.0)]$. Black line shows prediction of Eq. S81 (SI Sec. I.C). Data in (b), (c), and (d) denote discrete-time particle simulations.

$b_f = 1$ correspond to (noiseless) gradient descent (Fig. 3c). All results are qualitatively independent of particle volume fraction $\phi > \phi_c$, the potential $V_{ij}$, and the spatial dimension $d$, a useful feature since typical neural manifold dimensions are $\mathcal{O}(100)$[26] (SI Fig. S5).

Finally, we investigate the relationship between $\Delta E$ and the long-range structure formed by SGD. Combining the change in $S(k)$ before and after adding a small perturbation to the system[54] with our fluctuating hydrodynamic theory (Eq. (10)), we derive an expression for $\Delta E(c, b_f, \alpha)$ (SI Sec. I.C, Eqs. S79, S81, and S82)—in quantitative agreement with numerical simulations without free parameters (Fig. 3b–d). Thus, the emergent long-range structure in SGD provides a quantitative framework for understanding the flatness of energy landscape, a key feature linked to generalization. This unites two seemingly disparate domains—neural networks and interacting-particle systems—by revealing that the bias of SGD towards flat regions is a universal hallmark of high-dimensional loss (or energy) landscapes, regardless of the underlying system. Beyond its relevance to machine learning algorithms, our framework may guide the design of self-organizing materials with tunable energetic and structural properties.

To understand the characteristics of the flatter regions of energy landscape explored by SGD, we also measure the energy $E$ of the steady-state configurations $\mathbf{X}$ in particle simulations. The dependence of $E$ on key parameters—noise correlation $c$, batch fraction $b_f$, and learning rate $\alpha$—is opposite to that of the energy change for small perturbations $\Delta E$ (SI Figs. S5, and S6). This finding is in quantitative agreement with predictions from our fluctuating hydrodynamic theory (SI. Sec. I.C, Eqs. S72, S74, S75, and Fig. S6). Notably, the magnitude of these changes differs significantly: variation in system parameters leads to a substantial change in $\Delta E \sim 50\%$, but only a minor change in $E \sim 8\%$. Therefore, flatter regions of the energy landscape explored by SGD are associated with nearly constant or slightly higher energy. A similar trend is also observed in SGD dynamics on neural networks, where flatter regions of loss landscape often correlate with constant or slightly higher loss[30].

## Discussion

Random-organizing systems offer an ideal framework to probe noise-driven self-organization. Combining simulations and theory, we provide a unified, microscopic description of dynamics and emergent organization across a diverse array of random-organizing systems. We reveal that universal long-range behavior arises away from criticality in these microscopically distinct systems, dictated solely by interparticle

noise correlations. Finally, we demonstrate that SGD in particle systems inherently biases dynamics toward flatter energy regions, mirroring the behavior of SGD in neural networks—highlighting deep parallels between SGD dynamics in these two high-dimensional, but otherwise completely distinct, systems.

Random-organizing systems are inherently athermal (Eqs. 1, 2, and 3). We investigate how their long-range structure changes at a finite temperature by incorporating thermal noise, $\sqrt{2D^{th}}\,\xi_i^{th}(t)$, into the generalized model (Eq. 5), where $\xi_i^{th}(t)$ is a standard Gaussian white noise, $D^{th} = k_B T/\gamma$, $k_B$ is the Boltzmann constant, and $T$ is the temperature. We assume that the thermal noise and the intrinsic athermal noise, $\xi_{ij}(t)$, are uncorrelated (Eq. 5, and SI Eq. S83). We find, both analytically and in particle simulations, that the infinite wavelength density fluctuations, quantified by $S(k \to 0)$, increase with temperature, across all noise correlations $c$ and systems (SI Sec. I.D, Eq. S85, Fig. S7). Notably, systems with anti-correlated noise ($c = -1$), which are hyperuniform when athermal, show $S(k \to 0) \propto T$, illustrating how hyperuniformity is affected by thermal fluctuations (SI Fig. S7 inset). This behavior, also observed in other non-equilibrium hyperuniform systems[49], is similar to how thermal fluctuations weaken hyperuniformity in equilibrium crystals, as described by the fluctuation-compressibility relationship $S(\mathbf{k} = \mathbf{0}) = \kappa \rho k_B T$, where $\kappa$ is the isothermal compressibility[7,55].

We now discuss the conditions under which the linearization approximation of our fluctuating hydrodynamic theory remains valid (Eqs. 6, and 10). It is well known that such approximations hold for high density and soft overlap potentials[56–59]. Indeed, our linearized theory does not explain the long-range structure either in the absorbing phase ($\phi < \phi_c$) or at criticality ($\phi = \phi_c$). In the absorbing phase, the long-range structure remains unchanged by the noise correlation $c$ (SI Fig. S8). At criticality, the long-range structure is still independent of $c$ but becomes strongly dependent on dimension $d$: for $d < 4$ the systems are hyperuniform ($S(k \to 0) \sim k^\alpha$), whereas for $d \geq 4$ they are random ($S(k \to 0) \sim$ const.), with the exponent $\alpha$ varying with $d$[7,17,18,21,23,60,61].

From an application perspective, disordered hyperuniform systems possess features such as isotropic photonic bandgaps at low dielectric constants[62–64], defect-insensitive bandgaps[65], exceptional transparency[66], and high absorption rates in solar cells[67]. Consequently, designing structures with tunable hyperuniformity is highly desirable. Since many-body, and long-range interactions are difficult to realize in experiments, random-organizing systems offer a promising framework for designing hyperuniform structures due to their two-

body, and short-range interactions. Hyperuniformity at criticality in random-organizing systems, however, is weak (class III hyperuniformity[7]) and highly sensitive to density, with even slight density variations (~ 0.5%) capable of disrupting it[9,10,17,18,25]. In contrast, hyperuniformity in the active phase is strong (class I hyperuniformity[7]), and independent of density, providing a robust approach for experimental realization.

Finally, our study focuses on random-organizing systems with pairwise (two-body) interactions. How does the long-range structure change when many-body interactions are introduced in such systems? We hypothesize that imposing many-body interactions could give rise to finer control over the long-range structure, opening new avenues for designed self-assembly in complex materials.

## Methods
### Particle simulations
Our system consisted of $N$ hyperspherical particles of radius $R$ in a $d$-dimensional hypercubic box of side length $L$ with periodic boundary conditions. The unit of length, time, and energy were chosen as $2R$, $\delta t$, and $\mathcal{E}$, respectively. $\delta t$ is the simulation time-step for continuous-time simulations and $\mathcal{E} = 1$ is the characteristic energy scale of the potential given by Eq. (4). $N = 318309$ and $R = 1$ were kept fixed and the particle volume fraction $\phi = NV_s/V_c$ was varied by changing $L$. $V_s$ and $V_c$ denote volumes of a $d$-dimensional hypersphere of radius $R$, and hypercube of side length $L$, respectively. Particles were randomly distributed in the simulation box at $t = 0$. All simulations were run until the system reached a steady-state and all measurements were performed at steady-state. All results were averaged over 100 steady-state configurations.

**Discrete-time simulations.** For discrete-time simulations of RO, BRO, and SGD, the dynamics of particles were evolved according to Eqs. (1), (2), and (3), respectively. As evident from Eqs. (1), (2), and (3), positions of isolated particles (particles with no overlapping neighbors) do not evolve at any given time-step.

Dynamics for RO are controlled by four parameters, the kick magnitude $\epsilon$, the particle volume fraction $\phi$, the spatial dimension $d$, and the correlation of pairwise noise $c$. For the results reported in the main text, parameters were set as $\epsilon = 1$, $\phi = 1.0$, $d = 2$, and $c \in [-1, 0]$. The critical volume fraction $\phi_c \approx 0.375$ for this set of parameters.

Dynamics for BRO are controlled by four parameters, the kick magnitude $\epsilon$, the particle volume fraction $\phi$, the spatial dimension $d$, and the correlation of pairwise noise $c$. For the results reported in the main text, parameters were set as $\epsilon = 1$, $\phi = 1.0$, $d = 2$, and $c \in [-1, 0]$. The critical volume fraction $\phi_c \approx 0.395$ for this set of parameters.

Dynamics for SGD are controlled by six parameters, the learning rate $\alpha$, the particle volume fraction $\phi$, the batch fraction $b_f$, the "stiffness" of the potential $p$, the spatial dimension $d$, and the correlation of pairwise noise $c$. For the results reported in the main text, parameters were set as $\alpha = 0.5$, $\phi = 1.0$, $b_f = 0.5$, $p = 1$, $d = 2$, and $c \in [-1, 0]$. The critical volume fraction $\phi_c \approx 0.615$ for this set of parameters.

For the measurement of flatness of energy landscape in SGD, a configuration $\mathbf{X} \equiv \{\mathbf{x}_1, \mathbf{x}_2, \ldots, \mathbf{x}_N\}$ was perturbed by adding an independent Gaussian noise $\mathbf{N}_i(\mathbf{0}, \sigma^2\mathbf{I})$ to the position of each particle to get $\mathbf{X} + \Delta\mathbf{X} \equiv \{\mathbf{x}_1 + \mathbf{N}_1, \mathbf{x}_2 + \mathbf{N}_2, \ldots, \mathbf{x}_N + \mathbf{N}_N\}$. All results were averaged over 5000 independent noise realizations. For the results reported in Fig. 3b, parameters were set as $\alpha = 0.5$, $b_f = 0.5$, $\phi = 1.0$, $p = 1$, $d = 2$, $\sigma = 0.01$, and $c \in [-1, 0]$. For the results reported in Fig. 3c, parameters were set as $\alpha = 0.5$, $c = -1$, $\phi = 1.0$, $p = 1$, $d = 2$, $\sigma = 0.01$, and $b_f \in [0.1, 1.0]$. For the results reported in Fig. 3d, parameters were set as $b_f = 0.5$, $c = -1$, $\phi = 1.0$, $p = 1$, $d = 2$, $\sigma = 0.01$, and $\alpha \in [0.5, 1.0]$.

**Continuous-time simulations.** For the continuous-time simulations of the generalized model of RO, BRO, and SGD, the dynamics of particles were evolved according to Eq. (5), supplemented with appropriate values of the friction constant $\gamma$, and the short-range interaction potential $V_{ij}$ and matrix $\Lambda_{ij}$. The Euler-Maruyama method was used to solve Eq. (5), with a time-step $\delta t = 1.0$.

For the generalized model of RO, $V_{ji} = 0$, and $\Lambda_{ji,\alpha\beta} = \mathbf{1}_{(0, 2R)}(r_{ij})(\epsilon^2/3d\tau)\delta_{\alpha\beta}$, where $\mathbf{1}_{(0, 2R)}(r_{ij})$ is an indicator function such that $\mathbf{1}_{(0, 2R)}(r_{ij}) = 1 \,\forall\, r_{ij} \in (0, 2R)$ and $\mathbf{1}_{(0, 2R)}(r_{ij}) = 0$ otherwise. $\tau$ is the time scale quantifying the time elapsed in a discrete step. For the results reported in the main text, parameters were set as $\epsilon = 1$, $\tau = 1.0$, $\phi = 1.0$, $d = 2$, and $c \in [-1, 0]$.

For the generalized model of BRO, $\gamma = \mathcal{E}\tau/\epsilon R$, $\Lambda_{ji,\alpha\beta} = (\epsilon^2 R^2/3\tau\mathcal{E}^2)\,\partial_\alpha V_{ji}\partial_\beta V_{ji}$, and $V_{ji}$ is a short-range, pairwise, linear, repulsive potential (Eq. (4) with $p = 1$). For the results reported in the main text, parameters were set as $\epsilon = 1$, $\tau = 1.0$, $\phi = 1.0$, $d = 2$, and $c \in [-1, 0]$.

For the generalized model of SGD, $\gamma = \tau/\alpha b_f$, $\Lambda_{ji,\alpha\beta} = (\alpha^2 b_f(1 - b_f)/\tau)\,\partial_\alpha V_{ji}\partial_\beta V_{ji}$, and $V_{ji}$ is given by Eq. (4). For the results reported in the main text, parameters were set as $\alpha = 1$, $b_f = 0.5$, $\tau = 1$, $p = 1$, $\phi = 1.0$, $d = 2$, and $c \in [-1, 0]$.

### Continuum simulations
Our system consisted of density $\rho(\mathbf{x}, t)$ evolving in a $d$-dimensional hypercubic box of side length $L$ with periodic boundary conditions. The unit of length, time, and energy were chosen as $R$, $\delta t$, and $\mathcal{E}$, respectively. $\delta t$ is the simulation time-step, $\mathcal{E} = 1$ is the characteristic energy scale of the continuous version of the potential given by Eq. (4), and $2R$ is the cutoff length of the potential. Space was discretized with a square grid of grid spacing $\delta l = L/512$. $L = 200$ and $R = 1$ were kept fixed and $\int\rho(\mathbf{x}, t)d\mathbf{x} = N$ was fixed at all times (density conservation). Density at all grid points was sampled from a Gaussian distribution with mean $N/V_c$, and standard deviation $0.01N/V_c$ at $t = 0$. All other parameters for RO, BRO, and SGD were chosen to be the same as that of continuous-time particle simulations. All simulations were run until the system reached a steady-state and all measurements were performed at steady-state.

The finite-difference method combined with forward Euler time stepping was used to solve Eq. (6). Spatial derivatives for the stochastic flux and the diffusion term in Eq. (6) were approximated by the second-order central difference scheme. We split the velocity term (Eqs. (6), (7)) into two parts using chain rule,

$$\frac{1}{\gamma}\nabla \cdot \left[\rho(\mathbf{x})\nabla\langle V(\mathbf{x}, \mathbf{y})\rangle_{\rho(\mathbf{y})}\right] = \frac{1}{\gamma}\nabla\rho(\mathbf{x}) \cdot \nabla\langle V(\mathbf{x}, \mathbf{y})\rangle_{\rho(\mathbf{y})} + \frac{1}{\gamma}\rho(\mathbf{x})\,\nabla^2\langle V(\mathbf{x}, \mathbf{y})\rangle_{\rho(\mathbf{y})}. \tag{12}$$

Spatial derivatives in the first and second term on the right hand side of Eq. (12) were approximated using the first-order forward difference, and the second-order central difference scheme, respectively. The finite-difference schemes were chosen to ensure strict density conservation and prevent checkerboard artifacts.

### Structure factor and density fluctuations
The structure factor for a particle system is defined as $S(\mathbf{k}) = |\widehat{\rho}(\mathbf{k})|^2/N$, where $\rho(\mathbf{x}) = \sum_{i=1}^N \delta(\mathbf{x} - \mathbf{x}_i)$, and $\widehat{f}$ is the spatial Fourier transform of any arbitrary function $f(\mathbf{x})$, given by $\widehat{f}(\mathbf{k}) = \int f(\mathbf{x})e^{-i\mathbf{k}\cdot\mathbf{x}}d\mathbf{x}$. $S(\mathbf{k})$ was calculated using the nonuniform fast Fourier transform[68,69]. The radial structure factor $S(k)$ was then calculated by radially averaging $S(\mathbf{k})$.

The structure factor for a continuous density field $\rho(\mathbf{x})$ is defined as $S(\mathbf{k}) = |\widehat{\delta\rho}(\mathbf{k})|^2/\bar{\rho}$, where $\delta\rho(\mathbf{x}) = \rho(\mathbf{x}) - \bar{\rho}$, and $\bar{\rho} = \int \rho(\mathbf{x})d\mathbf{x}/\int d\mathbf{x}$.

Number density variance $\delta\rho^2(l)$ in a hyperspherical window of diameter $l$ was measured by exploiting the exact relationship between $S(k)$ and $\delta\rho^2(l)$[7],

$$\delta\rho^2(l) = \frac{\bar{\rho}d2^d\Gamma(1 + \frac{d}{2})}{(\sqrt{\pi}l)^d}\int_0^\infty \frac{1}{k}S(k)\left[J_{d/2}\left(\frac{kl}{2}\right)\right]^2 dk, \tag{13}$$

where $\bar{\rho} = N/V_c$, $\Gamma$ is the Gamma function, and $J$ is the Bessel function of the first kind. The integral in Eq. (13) was evaluated numerically using Simpson's rule, based on the measured $S(k)$.

## Data availability

The data generated in this study are provided in the article and its Supplementary Information file.

## Code availability

The code is available at https://github.com/guanming-zhang/sips for particle simulations and https://github.com/guanming-zhang/Bromf for continuum simulations. G.Z. implemented the base code, and S.A. implemented the extended features for pairwise correlations and random-organization dynamics.

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

## Acknowledgements

We thank Mathias Casiulis, David Heeger, Flaviano Morone, and Aaron Shih for fruitful discussions. This work was supported by the National Science Foundation grant IIS-2226387 and Award 2443027, National Institute of Health under award number R01MH137669, Simons Center for Computational Physical Chemistry, and in part by the NYU IT High Performance Computing resources, services, and staff expertise.

## Author contributions

S.A., G.Z. and S.M. conceived the project. S.A. and G.Z. performed the research. S.A., G.Z. and S.M. analyzed data and wrote the manuscript. S.M. supervised the research.

## Competing interests

The authors declare no competing interests.
