## [Transparent Peer Review file · Nature Communications]

Emergent universal long-range structure in random-organizing systems

Corresponding Author: Mr Satyam Anand

Version 0:

Reviewer comments:

Reviewer #1

(Remarks to the Author)

The authors study the suppression of long-range density fluctuations (i.e., hyperuniformity) in stochastic many-body dynamics. Specifically, they compare three dynamics: random organization (RO), biased random organization (BRO), and stochastic gradient descent (SGD) [Fig. 2]. They argue that these dynamics can be mapped into a generalized model [Eq. (5)]. Using analytical coarse-graining, they predict the emergence of a lengthscale [Eq.(11)] controlling the emergence of hyperuniformity [Fig. 2], and examine the flatness of energy minimum [Fig. 3].

The manuscript is an interesting addition to the existing literature on hyperuniformity. The main novelty stems from exploiting analogies between distinct stochastic dynamics that describe either soft matter systems or algorithms in machine learning. The analytical and numerical results are clearly non-trivial and carefully presented. My main concern regards the time-discretization convention of the multiplicative noise in the stochastic dynamics [Eq.(5)], see details below.

Overall, I recommend publication of the manuscript in Nature Communications, provided that the comments below are properly addressed.

1. The authors exploit analogies between SGD in particle systems and neural networks. Here, they should acknowledge previous works that have already examined such analogies, for instance [Comm. Pure Appl. Math 75, 1889 (2022)]. In the derivation of the fluctuating hydrodynamics, the authors should refer to other studies which have addressed the coarse-graining of stochastic dynamics with pairwise multiplicative noise; for instance [J Chem Phys 140, 234115 (2014)].
2. The authors should clarify the time-discretization convention of the stochastic dynamics with multiplicative noise [Eq.(5)] to lift any ambiguity regarding the associated Fokker-Planck equation. In fact, the numerical simulations of Eq.(5) should properly take into account this convention (e.g., Ito or Stratonovitch). Moreover, it would be useful to determine whether the steady-state distribution is actually given by the Boltzmann factor associated with the pairwise potential V_{ij} [Eq.(4)].
3. The authors should acknowledge that the stochastic dynamics of the empirical density [Eq.(6)] actually contains the same information as the many-body dynamics [Eq.(5)]. In other words, there is not any assumption (in particular, the two-point density is not assumed to be given by the product of empirical densities) in deriving the fluctuating hydrodynamic theory. It would be useful to clarify whether the functional steady-state distribution of the empirical density is given by the Boltzmann distribution associated with the functional free-energy (as expected for equilibrium dynamics).
4. The authors should clarify in which regime of parameters the linearization of density fluctuations that underlies the analytical form of the structure factor [Eq.(10)], should be valid. In fact, one expects that this regime corresponds to high densities and weak interactions; for instance, see previous works where a similar linearization has been used [New J. Phys. 16, 053032 (2014); Phys. Rev. X 9, 041026 (2019); Phys. Rev. Lett. 133, 268002 (2024); review: arXiv:2411.13467].

(Remarks on code availability)

Reviewer #2

(Remarks to the Author)

The main contribution of this work is the analysis of a range of microscopic models using simulations, continuous theory and coarse-graining that display in some part of their phase diagram a non-equilibrium steady state characterized by the fact that the explored configurations are hyperuniform. We find the paper very nice and timely, but would like to ask the author to take our remarks into account before recommending publication.

1. Some details of the model definitions are confusing. We understand the control of the degree of anticorrelation in the displacement components but the author does not explain how in practice they achieve this: do they pick one component, and then draw the second one as a sum of two terms weighted by c ? Please clarify.
2. A number of contributions could be cited much better: the work of Harukuni Ikeda who studied a range of models with special noise correlations in several publications should be acknowledged better. Also Ran Ni and coworkers and also Jack et al. (Phys. Rev. Lett. 114, 060601 2015) have developed coarse-grained analysis for hyperuniformity, please comment on possible links.
3. We are not convinced by the whole discussion of "flat minima". First, that part feels totally disconnected from the main point of the work, and the questions raised appear somewhat artificial. Second, we are unsure about some details of the analysis as the manuscript mentions that the small perturbations added to the system are in "steady state" configurations, which are not minima at all - can the authors specify more precisely what they do, minima of what they consider and how? In addition, the derived connection between $S(k)$ and ΔE relies on many approximations that are not clear to us: linearized Dean equation and low- k behavior of $S(k)$ whereas energy and interactions rather stem from large k (interparticle distance). The agreement displayed is suspiciously good. Anyhow, the causality between $S(k \rightarrow 0)$ and "flat minima" is largely overemphasized, as best a correlation holds, but given the above caveat, we are unsure. In summary: We recommend to remove that part entirely.
4. In the same vein, the dilute Dean equation is, strictly speaking, valid for dilute or weakly interacting systems. Please comment on this, and why would this approximation be valid for the present simulated systems is unclear (and indeed unreasonable).

(Remarks on code availability)

Reviewer #3

(Remarks to the Author)

In the manuscript by Anand, et al., the authors investigate the long-range correlated structures in random organising models (RO and BRO) with noise of different correlations, and they also tried to generalise it to the stochastic gradient descent (SGD), which is a commonly used method for machine learning. The results are interesting, but I have a few questions that need to be addressed before I could make any recommendation.

1. The central idea in this work is the influence of c , which is the correlation between the noise vectors acting on two active particles i and j . They found that the system is hyperuniform ($S(k) \sim k^{-2}$) until a length-scale, which increases with decreasing c and diverges at $c = -1$. Actually, this effect can be probably also seen as a combination of a noise conserving the center of mass and another (white) noise. Essentially, the effect studied here is probably just the effect of thermal noise or temperature on non-equilibrium hyperuniform fluids as in Ref [44]. In case the (white) noise is very strong, $c = 0$, which corresponds to the high temperature case. In [44], it was found that the effect of thermal noise (of strength T) can be decoupled from the noise conserving the center of mass of the system, and $S(k) \sim k^{-2}$ up to a length-scale, which diverges at the zero thermal noise limit ($T \rightarrow 0$). When $T \ll 1$, $S(0) \sim T$, which recovers Eq. 10 in this work. Therefore, I would suggest the authors to compare their theory and results with the effect of thermal noise as in Ref [44]. If they are essentially the same, the findings in this work may not be universal, as it may depend on the exact form, of which the noise is injected into the system, e.g., the colour and/or function form of (thermal) noise.
2. Following the question above, in [44], it was found that $S(0) \sim T$ is only valid at the low temperature region ($T \ll 1$), which is similar to the effect of thermal noise on equilibrium crystals at low temperature, and I think Eq. 10 may also have the similar constraint, which should be valid only when c is near -1 . Therefore, I would suggest the authors to clarify this.
3. I don't understand the difference between RO and BRO as shown in Fig. 2, as my understanding is that both models are the same at the fixed c , in which the noise vectors have the same correlation c between i and j . I would suggest the authors to clarify the difference between the two models, if there are any.

(Remarks on code availability)

Version 1:

Reviewer comments:

Reviewer #1

(Remarks to the Author)

The authors have significantly changed the main text and the supplementary information to address the issues raised by the referees. I still have some comments, as detailed below. Should these comments be properly addressed by the authors, I would recommend the manuscript for publication at Nature Communications.

1. In section 'Fluctuating hydrodynamic theory' of the main text, the authors discuss some previous works on coarse-graining stochastic dynamics with multiplicative noise, and mention

'However, it has not yet been extended to systems with pairwise correlated noise between components'

As far as I understand, Ref.[48] actually addresses the case of Brownian particles with a multiplicative pairwise noise. It would be useful if the authors clarified to which extent their derivation actually differs from that of Ref.[48]

2. In the supplementary information, the authors demonstrate that their dynamics follows a Boltzmann steady-state measure at equilibrium, as it should. Here, equilibrium corresponds to choosing a specific noise statistics that amounts to enforcing some additive, diagonal noise correlations. In other words, the three models considered by the authors are generally out-of-equilibrium. Although 'Random Organization' and 'Biased Random Organization' are clearly inspired by nonequilibrium systems, it could be useful to comment further on why 'Stochastic Gradient Descent' should be regarded as a nonequilibrium dynamics.

In section 'Stochastic Gradient Descent' of the main text, the authors mention that SGD is meant to minimize the total energy. It would be insightful to show that the steady-state distribution of the stochastic dynamics in Eq.(5) is indeed highest at the minimum energy of the system. In other words, the authors should demonstrate that such a property holds for the specific choice of diffusion matrix in table S1.

(Remarks on code availability)

Reviewer #2

(Remarks to the Author)

I have reviewed the response and revisions of the author to the comments made in the first round and I am satisfied by their work. I can recommend publication at this point.

(Remarks on code availability)

Reviewer #3

(Remarks to the Author)

I appreciate the efforts that the authors have spent in revising the manuscript and the explanation they have offered to the questions in my previous report. I agree with the authors that the influence of c is not exactly the same as the effect of thermal noise in random organising hyperuniform fluids in the literature, as they come with slightly different mathematical forms. But I think the essential physics in the effect of c is the same as adding an extra noise, and in this work it is a white noise but not exactly the same way as being added in the active hyperuniform fluids previously. I believe that the effect of c can be also decoupled as an extra noise term in addition to the noise conserving the center of the mass of the system. Then the finding in this work is not "universal" as claimed by the authors, as the effect should depend on the exact form of the extra noise, such as the color of the noise, the mathematical form of correlation in the noise, etc. Therefore, I don't think this manuscript is suitable for publications in high profile journals like Nature Communications, but rather suggest the authors to resubmit it (with revision) to more specialised journals.

(Remarks on code availability)

Version 2:

Reviewer comments:

Reviewer #1

(Remarks to the Author)

The authors have provided satisfactory answers to all my comments. Therefore, I recommend publication of the manuscript in its present form.

(Remarks on code availability)

Reviewer #3

(Remarks to the Author)

I have read the response from the authors, which I disagree with. They insisted that there is only one noise in the system controlling but c with $c=0$ being uncorrelated and $c=-1$ anti-correlated, but as I mentioned that it can always be decomposed into two noises with one anti-correlated responsible for the hyperuniformity and the other one responsible for deviation from hyperuniformity. This picture actually agrees with what they have seen in their simulation, which is hyperuniform to certain lengthscale and can be seen as a distorted hyperuniformity. Therefore, I think that the authors overrated the significance of this work, and I do not think that the phenomena they saw is universal, which should depend on the exact form of the other noise changing the hyperuniformity of the system. So I cannot recommend for publication in Nature Communications.

(Remarks on code availability)

Response to the reviewers' report

We report below a detailed response to the reviewers' reports. The original reviewer comments are in italic, and our response appears in normal font. When addressing the reviewers' comments, we also explicitly list the changes we made in the revised manuscript and highlight those changes in red here and in the revised manuscript.

Response to Reviewer 1

We thank the reviewer for their positive comment and their recommendation for our work to be published. Below we address the questions raised by the reviewer.

1. The authors exploit analogies between SGD in particle systems and neural networks. Here, they should acknowledge previous works that have already examined such analogies, for instance [Comm. Pure Appl. Math 75, 1889 (2022)]. In the derivation of the fluctuating hydrodynamics, the authors should refer to other studies which have addressed the coarse-graining of stochastic dynamics with pairwise multiplicative noise; for instance [J Chem Phys 140, 234115 (2014)].

We thank the reviewer for pointing us to the relevant literature. Following the suggestions of the reviewer, we have added both references in the main text. The modified ‘‘Stochastic Gradient Descent’’ section of the main text including reference [Comm. Pure Appl. Math 75, 1889 (2022)] now reads - ‘‘**Notice that our approach of SGD applied on particle systems is unlike previous works relating neural networks to particle systems by treating parameters (weights) as interacting particles [1].**’’

Further, the modified ‘‘Fluctuating hydrodynamic theory’’ section of the main text including reference [J Chem Phys 140, 234115 (2014)] now reads - ‘‘**Dean’s method, originally introduced for Brownian particles with additive noise [2], and later extended to study systems with multiplicative noise [3–5], is a well-known approach for directly coarse-graining microscopic dynamics.**’’

2. The authors should clarify the time-discretization convention of the stochastic dynamics with multiplicative noise [Eq.(5)] to lift any ambiguity regarding the associated Fokker-Planck equation. In fact, the numerical simulations of Eq.(5) should properly take into account this convention (e.g., Ito or Stratonovitch). Moreover, it would be useful to determine whether the steady-state distribution is actually given by the Boltzmann factor associated with the pairwise potential V_{ij} [Eq.(4)].

We thank the reviewer for pointing out about the time-discretization convention. Eq.(5) in the main text is interpreted in the Itô sense. We have added the phrase: ‘‘... and Eq. 5 is interpreted in the Itô sense’’, after Eq.5 in the main text. For the numerical simulations of Eq.5, we used the Euler–Maruyama method, which we had mentioned in the Methods section ‘‘Continuous-time simulations’’ of the main text.

Further, in the SI, we show that since random-organizing systems are out-of-equilibrium, the steady-state distribution for the N -body Fokker-Planck equation associated with Eq.(5) is not given by the Boltzmann distribution. We have added the following discussion in SI. Sec. I.B.1. -

Steady-state distribution. Here we address the question of whether the steady-state solution for the evolution of the N -particle density ρ_N is given by the Boltzmann distribution, as for interacting Brownian particles at equilibrium [6, 7] (Eq. S16). Plugging the steady-state distribution ρ_N^s in Eq. S16 and noting that $\partial \rho_N^s / \partial t = 0$, we get,

$$\frac{1}{\gamma} \sum_i \partial_{x_i, \alpha} \left[\left(\partial_{x_i, \alpha} \sum_j V_{ji} \right) \rho_N^s \right] + \frac{1}{2} \sum_i \sum_j \partial_{x_i, \alpha} \partial_{x_j, \beta} \left[\left(\delta_{ij} \sum_k \Lambda_{ki, \alpha\beta}(t) + c \Lambda_{ij, \alpha\beta} \right) \rho_N^s \right] = 0. \quad (\text{R1})$$

Notice that Eq. R1 is difficult to solve for arbitrary $\Lambda_{ij, \alpha\beta}$ and noise correlation c . However, substituting $\Lambda_{ij, \alpha\beta} = a \delta_{\alpha\beta}$, and $c = 0$ in Eq. R1, where a is some constant, we get after some algebra,

$$\sum_i \partial_{x_i, \alpha} \left[\frac{1}{\gamma} (\partial_{x_i, \alpha} U_i) \rho_N^s + \frac{Na}{2} \partial_{x_i, \alpha} \rho_N^s \right] = 0, \quad (\text{R2})$$

where $U_i = \sum_j V_{ij}$, and N is the number of particles. The solution of Eq. R2 is given by the Boltzmann distribution $\rho_N^B = (1/Z)e^{-(2/Na\gamma)E}$, where $E = \sum_i \sum_{j \geq i} V_{ij}$, and Z is the normalization factor. This is not surprising since substituting $\Lambda_{ij,\alpha\beta} = a\delta_{\alpha\beta}$ and $c = 0$ in Eq. S13 reduces it to a system of interacting Brownian particles at equilibrium, whose steady-state solution is known to be the Boltzmann distribution [6, 7]. Notice, however, that since $\Lambda_{ij,\alpha\beta} \neq a\delta_{\alpha\beta}$ and $c \neq 0$, in general, for random-organizing systems (Table S1), the Boltzmann distribution is not the steady-state solution for Eq. R1.

3. *The authors should acknowledge that the stochastic dynamics of the empirical density [Eq.(6)] actually contains the same information as the many-body dynamics [Eq.(5)]. In other words, there is not any assumption (in particular, the two-point density is not assumed to be given by the product of empirical densities) in deriving the fluctuating hydrodynamic theory. It would be useful to clarify whether the functional steady-state distribution of the empirical density is given by the Boltzmann distribution associated with the functional free-energy (as expected for equilibrium dynamics).*

We agree with the reviewer that the stochastic dynamics derived using Dean’s approach formally contains the same information as the many-body dynamics. To reflect this, we have removed the assumption about the two-point density being given by the product of empirical densities from the main text. The modified “Fluctuating hydrodynamic theory” section of the main text now reads - “Starting from Eq. 5, we extend Dean’s method [2] to account for multiplicative, and pairwise correlated noise, and derive the time evolution of $\rho(\mathbf{x}, t)$ in arbitrary dimension d to get ...”

We have also added in the SI Sec. I.B.2 - “Note that taking the ensemble average (over noise realizations) of Eq. S42, and using the mean-field approximation $\rho_2(\mathbf{x}_1, \mathbf{x}_2) = \rho_1(\mathbf{x}_1)\rho_1(\mathbf{x}_2)$, we recover the density evolution equation derived previously using the Fokker-Planck method (Eq. S19, Sec. I.B.I)”

Further, we show in the SI that the functional Fokker-Planck equation associated with the stochastic density evolution (Eq. 6) does not have the Boltzmann density functional as its solution, since random-organizing systems are at a non-equilibrium steady-state. We have added the following discussion in SI. Sec. I.B.2. -

Functional steady-state distribution. The functional Fokker-Planck equation for Eq. S42 is given by,

$$\begin{aligned} \frac{\partial P[\rho, t]}{\partial t} = \int d\mathbf{x} \frac{\delta}{\delta \rho(\mathbf{x})} \left\{ - \left[\frac{1}{\gamma} \partial_{x,\alpha} \left(\rho(\mathbf{x}) \int \rho(\mathbf{y}) \partial_{x,\alpha} V(\mathbf{x}, \mathbf{y}) d\mathbf{y} \right) + \frac{1}{2} \partial_{x,\alpha} \partial_{x,\beta} \left(\rho(\mathbf{x}) \int \rho(\mathbf{y}) \Lambda_{\alpha\beta}(\mathbf{x}, \mathbf{y}) d\mathbf{y} \right) \right] P \right. \\ \left. + \frac{1}{2} \int d\mathbf{y} \frac{\delta}{\delta \rho(\mathbf{y})} \left(\left[\partial_{x,\alpha} \partial_{y,\mu} \left(\delta(\mathbf{x} - \mathbf{y}) \rho(\mathbf{x}) \int \rho(\mathbf{u}) \Lambda(\mathbf{x}, \mathbf{u})_{\alpha\mu} d\mathbf{u} + c \rho(\mathbf{x}) \rho(\mathbf{y}) \Lambda(\mathbf{x}, \mathbf{y})_{\alpha\mu} \right) \right] P \right) \right\}, \quad (\text{R3}) \end{aligned}$$

where $P[\rho, t]$ is the probability distribution functional of the density $\rho(\mathbf{x}, t)$. Plugging the steady-state distribution functional P^s in Eq. R3 and noting that $\partial P^s / \partial t = 0$, we get,

$$\begin{aligned} \int d\mathbf{x} \frac{\delta}{\delta \rho(\mathbf{x})} \left\{ - \left[\frac{1}{\gamma} \partial_{x,\alpha} \left(\rho(\mathbf{x}) \int \rho(\mathbf{y}) \partial_{x,\alpha} V(\mathbf{x}, \mathbf{y}) d\mathbf{y} \right) + \frac{1}{2} \partial_{x,\alpha} \partial_{x,\beta} \left(\rho(\mathbf{x}) \int \rho(\mathbf{y}) \Lambda_{\alpha\beta}(\mathbf{x}, \mathbf{y}) d\mathbf{y} \right) \right] P^s \right. \\ \left. + \frac{1}{2} \int d\mathbf{y} \frac{\delta}{\delta \rho(\mathbf{y})} \left(\left[\partial_{x,\alpha} \partial_{y,\mu} \left(\delta(\mathbf{x} - \mathbf{y}) \rho(\mathbf{x}) \int \rho(\mathbf{u}) \Lambda(\mathbf{x}, \mathbf{u})_{\alpha\mu} d\mathbf{u} + c \rho(\mathbf{x}) \rho(\mathbf{y}) \Lambda(\mathbf{x}, \mathbf{y})_{\alpha\mu} \right) \right] P^s \right) \right\} = 0. \quad (\text{R4}) \end{aligned}$$

Eq. R4 is challenging to solve for general $\Lambda(\mathbf{x}, \mathbf{y})_{\alpha\beta}$ and c , and no closed-form solution is known. However, substituting $\Lambda(\mathbf{x}, \mathbf{y})_{\alpha\beta} = a\delta_{\alpha\beta}$, and $c = 0$ in Eq. R4, where a is some constant, we get,

$$\begin{aligned} \int d\mathbf{x} \frac{\delta}{\delta \rho(\mathbf{x})} \left\{ - \left[\frac{1}{\gamma} \partial_{x,\alpha} \left(\rho(\mathbf{x}) \int \rho(\mathbf{y}) \partial_{x,\alpha} V(\mathbf{x}, \mathbf{y}) d\mathbf{y} \right) + \frac{aN}{2} \partial_{x,\alpha} \partial_{x,\alpha} \rho(\mathbf{x}) \right] P^s \right. \\ \left. + \frac{aN}{2} \int d\mathbf{y} \frac{\delta}{\delta \rho(\mathbf{y})} \left(\left[\partial_{x,\beta} \partial_{y,\beta} (\rho(\mathbf{x}) \delta(\mathbf{x} - \mathbf{y})) \right] P^s \right) \right\} = 0, \quad (\text{R5}) \end{aligned}$$

where $\int \rho(\mathbf{x}) d\mathbf{x} = N$. Using integration by parts, and the fact that $\partial_{x,\alpha} f(\mathbf{x}, \mathbf{y}) = -\partial_{y,\alpha} f(\mathbf{x}, \mathbf{y})$ for any function f

which depends only on $|\mathbf{x} - \mathbf{y}|$, we get,

$$\int d\mathbf{x} \frac{\delta}{\delta \rho(\mathbf{x})} \left\{ \left[\frac{1}{\gamma} \partial_{x,\alpha} \left(\rho(\mathbf{x}) \int \rho(\mathbf{y}) \partial_{x,\alpha} V(\mathbf{x}, \mathbf{y}) d\mathbf{y} \right) + \frac{aN}{2} \partial_{x,\alpha} \partial_{x,\alpha} \rho(\mathbf{x}) \right] P^s + \frac{aN}{2} \partial_{x,\alpha} \left[\rho(\mathbf{x}) \partial_{x,\alpha} \left(\frac{\delta P^s}{\delta \rho(\mathbf{x})} \right) \right] \right\} = 0 \quad (\text{R6})$$

The solution of Eq. R6 is given by the Boltzmann distribution functional $P^s = P^B[\rho] \equiv (1/Z) e^{-(1/Na\gamma)F[\rho]}$, where Z is a constant, and $F[\rho]$ is the free energy functional given by,

$$F[\rho] = Na\gamma \int \rho(\mathbf{x}) (\ln \rho(\mathbf{x}) - 1) d\mathbf{x} + \iint \rho(\mathbf{x}) \rho(\mathbf{y}) V(\mathbf{x}, \mathbf{y}) d\mathbf{x} d\mathbf{y}. \quad (\text{R7})$$

This can also be seen by the fact that substituting $\Lambda(\mathbf{x}, \mathbf{y})_{\alpha\beta} = a\delta_{\alpha\beta}$, and $c = 0$ in Eq. S42 reduces it to a standard Dean's equation for a system of interacting Brownian particles at equilibrium, whose steady-state solution is known to be the Boltzmann distribution functional [2, 8]. However, since $\Lambda(\mathbf{x}, \mathbf{y})_{\alpha\beta} \neq a\delta_{\alpha\beta}$, and $c \neq 0$, in general, for random-organizing systems (Table S1), the Boltzmann distribution functional is not the steady-state solution for Eq. S42.

FIG. R1: Long-range structure in the absorbing phase ($\phi < \phi_c$) of random-organizing systems, where ϕ_c denotes the critical particle volume fraction. Radially averaged structure factor $S(k)$ versus dimensionless radial wave number $k \times (2R/2\pi)$ in discrete-time particle simulations, where R is the particle radius, for (a) random-organization (RO); (b) biased random-organization (BRO); and (c) stochastic gradient descent (SGD). All parameters are the same as in the main text (see Methods), except ϕ , which is set to $\phi/\phi_c = 0.9$ for all systems.

4. *The authors should clarify in which regime of parameters the linearization of density fluctuations that underlies the analytical form of the structure factor [Eq.(10)], should be valid. In fact, one expects that this regime corresponds to high densities and weak interactions; for instance, see previous works where a similar linearization has been used [New J. Phys. 16, 053032 (2014); Phys. Rev. X 9, 041026 (2019); Phys. Rev. Lett. 133, 268002 (2024); review: arXiv:2411.13467].*

We agree with the reviewer that our linearized fluctuating hydrodynamic theory is valid for high densities and weak interactions, as is well known for linearized Dean's-type hydrodynamics theories. We have performed particle simulations in the absorbing phase ($\phi < \phi_c$), which show that our theory is not valid in that regime (Fig. R1). We have also cited previous works which show the long-range structure at the critical point ($\phi = \phi_c$) is also different than what is predicted by our theory. We have now added the following paragraph in the "Discussion" section of the main text -

Under what conditions is the linearization approximation of our fluctuating hydrodynamic theory valid (Eqs. 6, and 10)? It is well known that such linearization approximations are valid for high density and soft overlap potentials [8–11]. Indeed, our linearized theory does not explain the long-range structure either in the absorbing phase ($\phi < \phi_c$) or at criticality ($\phi = \phi_c$). In the absorbing phase, the long-range structure remains unchanged by the noise correlation c (SI Fig. S8). At criticality, the long-range structure is still independent of c but becomes strongly dependent on dimension d : for $d < 4$ the systems are hyperuniform ($S(k \rightarrow 0) \sim k^\alpha$), whereas for $d \geq 4$ they are random ($S(k \rightarrow 0) \sim \text{const.}$), with the exponent α varying with d [12–18].

Response to Reviewer 2

We thank the reviewer for their positive comments. Below we address the questions of the reviewer point-by-point.

1. *Some details of the model definitions are confusing. We understand the control of the degree of anticorrelation in the displacement components but the author do not explain how in practice they achieve this: do they pick one component, and then draw the second one as a sum of two terms weighted by c ? Please clarify.*

We apologize for the confusion. The degree of anti-correlation in RO, BRO, and SGD is controlled solely by scalar random variables.

In RO, a random vector is chosen from the surface of a unit-hypersphere which denotes the kick direction from particle i to j , and the kick direction from particle j to i is always the opposite of that. The kick magnitudes for a pair of particles are drawn from a uniform distribution such that they are correlated with a correlation coefficient $-c$. Then, since the direction of kicks is always opposite, the correlation coefficient of the complete noise vector is c .

For BRO, the kick direction is deterministic and always opposite for a pair of particles. The kick magnitude is correlated in a similar way as RO giving rise to the complete noise vector being correlated with Pearson correlation coefficient c .

For SGD, the kick direction and magnitude are both deterministic. The selection, determined by a Bernoulli random variable, is correlated for a pair of particles, and correlated Bernoulli random variables are drawn for each pair.

To clarify this, we have added the following line in the section “Biased Random Organization” in the main text - “Notice that $c = -1$ corresponds to $u_{ij}^m = u_{ji}^m$, since $\hat{\mathbf{x}}_{ji}^m = -\hat{\mathbf{x}}_{ij}^m$.”. Further, we have also added the following line in the section “Stochastic Gradient Descent” in the main text - Notice that since $\nabla_i V_{ji}^m = -\nabla_j V_{ji}^m$, $c = -1$ corresponds to $\theta_{ij}^m = \theta_{ji}^m$.

2. *A number of contributions could be cited much better: the work of Harukuni Ikeda who studied a range of models with special noise correlations in several publication should be acknowledged better. Also Ran Ni and coworkers and also Jack et al. (Phys. Rev. Lett. 114, 060601 2015) have developed coarse-grained analysis for hyperuniformity, please comment on possible link.*

We thank the reviewer for pointing us to additional literature. Following the suggestions of the reviewer, we have cited two relevant works of Harukuni Ikeda in the main text section “Fluctuating hydrodynamic theory” - “First, in a variety of other systems having local center of mass conserving dynamics [19–23] (equivalent to $c = -1$ in random-organizing systems) and displaying hyperuniformity in the dense phase, the coarse-grained noise term is of Laplacian form $\nabla^2[\sqrt{\rho(\mathbf{x})}\omega(\mathbf{x})]$, where $\omega(\mathbf{x})$ is a spatially uncorrelated, one-point Gaussian noise field.”. Further, we have also cited Jack et al. (Phys. Rev. Lett. 114, 060601 2015) in the “Introduction” of the main text - “Away from criticality, in equilibrium, hyperuniformity requires long-range interactions [18], whereas out of equilibrium, it can emerge from long- or short-range, and even noisy interactions [19, 24–28].”. We had already cited three relevant works of Ran Ni and co-workers in the main text, which are, Lei et al., PNAS 116, 22983 (2019), Lei et al., J. Chem. Phys. 159 (2023), and Lei et al., Science advances 5, eaau7423 (2019).

While these studies all perform coarse-grained analyses of hyperuniformity, they differ from our work in both the models they consider and in their scope. Each examines a particular non-equilibrium system, uncovers hyperuniform scaling numerically, and then develops a corresponding continuum theory, phenomenological or microscopically derived. Yet none tackles random-organizing systems or offers a theory—microscopic or phenomenological—that explains the universal onset of hyperuniformity in the active phase across models. Moreover, they restrict themselves to strictly hyperuniform regimes and provide no tunable microscopic parameter (for example, a noise-correlation coefficient c) that smoothly bridges random and hyperuniform behavior within a single framework.

3. *We are not convinced by the whole discussion of “flat minima”. First, that part feels totally disconnected from the main point of the work, and the questions raised appear somewhat artificial. Second, we are unsure about some details of the analysis as the manuscript mentions that the small perturbations added to the system are in “steady state” configurations, which are not minima at all - can the authors specify more precisely what they do, minima of what they consider and how? In addition, the derived connection between $S(k)$ and ΔE relies on many approximations that are not clear to us: linearized Dean equation and low- k behavior of $S(k)$ whereas energy and interactions rather stem from large k (interparticle distance). The agreement displayed is suspiciously good. Anyhow, the causality between*

$S(k \rightarrow 0)$ and “flat minima” is largely overemphasized, as best a correlation holds, but given the above caveat, we are unsure. In summary: We recommend to remove that part entirely.

We appreciate the reviewer’s careful reading and have revised the “flat minima” section to address the reviewer’s concerns, but we have chosen to retain the section because, as we argue below, we believe it is relevant.

Our work focuses on the universal long-range structure exhibited by random-organizing systems in their active phase. Since SGD is a unique random-organizing system that has a well-defined energy (in contrast to RO and BRO which have no *explicit* energy, see Eqs. 1, 2, and 3 in the main text), exploring the connection between the energy-landscape perspective—unique to SGD—and the measured structure factor—common across all random organizing systems—is both natural and insightful. Moreover, since SGD is widely used in neural networks, which are also high-dimensional, it is worth asking whether the fact that SGD leads to flat regions of the landscape in neural networks also holds true for particle systems, which are completely distinct from neural networks. Finally, since the structure and energetic properties are intertwined for particle systems, it is relevant to discuss them in the same study, rather than separately. This strong connection is evidenced by the following *exact* relationship between energy E and $S(k)$ [29, 30],

$$E = \frac{\rho N}{2} \int V(\mathbf{r}) d\mathbf{r} + \frac{N}{2(2\pi)^d} \int \hat{V}(\mathbf{k}) (S(\mathbf{k}) - 1) d\mathbf{k}. \quad (\text{R8})$$

where $V(\mathbf{r}) = V(|\mathbf{r}_j - \mathbf{r}_i|)$ is the pairwise potential, ρ is the number density, and N is the total number of particles.

We agree with the reviewer that the use of term “flat minima” (borrowed directly from the deep learning literature [31–33]) is misleading since the configurations found by SGD when the system relaxes to steady-state are not necessarily minima of the potential energy surface. Hence, we have replaced the word “flat minima” by “flat regions of energy landscape” throughout the main text to remove any ambiguity.

The relationship between ΔE and $S(k)$ derived in the SI Sec. I.C does indeed rely on two main assumptions: (i) linearized Dean’s equation used to derive the $S(k)$; (ii) the low- k behavior of $S(k)$ instead of the full $S(k)$ (Eq. 10 in the main text). There are two reasons for this. First, for short-range potentials, $\hat{V}(\mathbf{k})$ is typically dominated by the small- k region; hence we can approximate $S(\mathbf{k})$ in Eq. R8 by the small- k (long-range) behavior. Second, while the reviewer correctly points out that energy should be dictated by large- k behavior, we emphasize that since $\int S(k) dk = 1$, the low- and high- k behavior of $S(k)$ are not *independent* of each other. In other words, the change in low- k behavior is also necessarily reflected in a change in the average number and/or overlap of particles. To further make our argument concrete, we also measure the energy E of the steady-state configurations in particle simulations, and find that E can also be quantitatively related to $S(k)$ through our theory (see SI. Sec. I.C for derivation, Eqs. S68, S71, S70, and Fig. R2). This reinforces the fact that the prediction of ΔE from theory is not a coincidence. In fact, other properties like the total energy E , can also be reliably predicted from our theory. We have now added the following paragraph in the main text in the section “Flatness of energy landscape in stochastic gradient descent”:

To understand the characteristics of the flatter regions of energy landscape explored by SGD, we also measure the energy E of the steady-state configurations \mathbf{X} in particle simulations. The dependence of E on key parameters—noise correlation c , batch fraction b_f , and learning rate α —is opposite to that of the energy change for small perturbations ΔE (SI Figs. S5, and S6). This finding is in quantitative agreement with predictions from our fluctuating hydrodynamic theory (SI. Sec. I.C, Eqs. S68, S71, S70, and Fig. S6). Notably, the magnitude of these changes differs significantly: variation in system parameters leads to a substantial change in $\Delta E \sim 50\%$, but only a minor change in $E \sim 8\%$. Therefore, flatter regions of the energy landscape explored by SGD are associated with nearly constant or slightly higher energy. A similar trend is also observed in SGD dynamics on neural networks, where flatter regions of loss landscape often correlate with constant or slightly higher loss [31].

4. In the same vein, the dilute Dean equation is, strictly speaking, valid for dilute or weakly interacting systems. Please comment on this, and why would this approximation be valid for the present simulated systems is unclear (and indeed unreasonable).

Our linearized fluctuating hydrodynamic theory is valid for high densities and weak interactions, as is well known for linearized Dean’s-type hydrodynamics theories [8–11]. We have performed particle simulations in the absorbing phase ($\phi < \phi_c$), which show that our theory is not valid in that regime (Fig. R1). We have also cited previous works which show that the long-range structure at the critical point ($\phi = \phi_c$) is also different than what is predicted by our theory. We have now added the following paragraph in the “Discussion” section of the main text:

Under what conditions is the linearization approximation of our fluctuating hydrodynamic theory valid (Eqs. 6, and

FIG. R2: Dependence of energy of the steady-state configuration on system parameters for Stochastic Gradient Descent (SGD). Normalized energy $\tilde{E}(c)$ versus noise correlation c for different particle volume fraction ϕ (a), spatial dimension d (b), and stiffness of the potential p (Eq. S11) (c). $E(c)$ is normalized as: $\tilde{E}(c) = [E(c) - E(c=0)]/[E(c=-1) - E(c=0)]$. Black lines show prediction of Eq. S68 (SI Sec. I.C). Normalized energy $\tilde{E}(b_f)$ versus batch fraction b_f for different particle volume fraction ϕ (d), spatial dimension d (e), and stiffness of the potential p (Eq. S11) (f). $E(b_f)$ is normalized as: $\tilde{E}(b_f) = [E(b_f) - E(b_f=1.0)]/[E(b_f=0.1) - E(b_f=1.0)]$. Black lines show prediction of Eq. S71 (SI Sec. I.C). Normalized energy change $\tilde{E}(\alpha)$ versus learning rate α for different particle volume fraction ϕ (a), spatial dimension d (b), and stiffness of the potential p (Eq. S11) (c). $E(\alpha)$ is normalized as: $\tilde{E}(\alpha) = [E(\alpha) - E(\alpha=0.5)]/[E(\alpha=1.0) - E(\alpha=0.5)]$. Black lines show prediction of Eq. S70 (SI Sec. I.C). All parameters are the same as Fig. S5 (see caption). All symbols denote discrete-time particle simulations.

10)? It is well known that such linearization approximations are valid for high density and soft overlap potentials [8–11]. Indeed, our linearized theory does not explain the long-range structure either in the absorbing phase ($\phi < \phi_c$) or at criticality ($\phi = \phi_c$). In the absorbing phase, the long-range structure remains unchanged by the noise correlation c (SI Fig. S8). At criticality, the long-range structure is still independent of c but becomes strongly dependent on dimension d : for $d < 4$ the systems are hyperuniform ($S(k \rightarrow 0) \sim k^\alpha$), whereas for $d \geq 4$ they are random ($S(k \rightarrow 0) \sim \text{const.}$), with the exponent α varying with d [12–18].

Response to Reviewer 3

We thank the reviewer for their comments. Below we address the questions of the reviewer point-by-point.

1. The central idea in this work is the influence of c , which is the correlation between the noise vectors acting on two active particles i and j . They found that the system is hyperuniform ($S(k) \sim k^2$) until a length-scale, which increases with decreasing c and diverges at $c = -1$. Actually, this effect can be probably also seen as a combination of a noise

FIG. R3: Long-range structure in the presence of thermal noise in random-organizing systems. Normalized radially averaged structure factor $\tilde{S}(k)$ versus normalized radial wave number \tilde{k} for random-organization (RO) (a), biased random-organization (BRO) (b), and stochastic gradient descent (SGD) (c) in discrete-time particle simulations. All parameters are the same as Fig. 2 in main text (see Methods). $\tilde{S} = S(k)/S_0(2\pi/L)$ where $S_0(2\pi/L)$ is the structure factor for $c = 0$ at $k = 2\pi/L$, and L is the side length of the simulation box. $\tilde{k} = k/k_0$ where k_0 is the value at which $\tilde{S}(k_0) = 1$ for anti-correlated noise ($c = -1$) of the same system. The solid colored lines show a combined best fit of Eq. S81 to various noise correlations c for each system, using D^{th} as the single fitting parameter. Insets show $\tilde{S}(\tilde{k} \rightarrow 0)$ versus thermal diffusion coefficient D^{th} . $\hat{S}(\tilde{k} \rightarrow 0, c, D^{\text{th}}) = \tilde{S}(\tilde{k} \rightarrow 0, c, D^{\text{th}})/\tilde{S}(\tilde{k} \rightarrow 0, c, 0, D^{\text{th}} = 0)$. Solid black lines in the insets denote $1 + c + 2D^{\text{th}}/\bar{\rho}A_1$ (prediction of Eq. S81 in the limit $\tilde{k} \rightarrow 0$). Gray shaded regions denote short-range behavior ($\tilde{k} > 1$).

conserving the center of mass and another (white) noise. Essentially, the effect studied here is probably just the effect of thermal noise or temperature on non-equilibrium hyperuniform fluids as in Ref [44]. In case the (white) noise is very strong, $c = 0$, which corresponds to the high temperature case. In [44], it was found that the effect of thermal noise (of strength T) can be decoupled from the noise conserving the center of mass of the system, and $S(k) \sim k^2$ up to a length-scale, which diverges at the zero thermal noise limit ($T \sim 0$). When $T \ll 1$, $S(0) \sim T$, which recovers Eq. 10 in this work. Therefore, I would suggest the authors to compare their theory and results with the effect of thermal noise as in Ref [44]. If they are essentially the same, the findings in this work may not be universal, as it may depend on the exact form, of which the noise is injected into the system, e.g., the colour and/or function form of (thermal) noise.

Following the suggestions of the reviewer, we investigated the effect of thermal noise on random-organizing systems, which are otherwise inherently athermal. We find, both analytically and in particle simulations, that, similar to ref. [20], the addition of thermal noise changes the $S(k \rightarrow 0)$ behavior in all systems, with a constant factor that increases with temperature T , in particular, $S(k \rightarrow 0) \propto 1 + c + aT$, where a is a system-dependent constant (see SI Sec.I.D. for derivation, Eq. S81). This is similar to how hyperuniformity is destroyed in ref. [20], and also in equilibrium crystals. In our work, we propose a more general and fundamental mechanism. We do not have two competing noise sources. Instead, we have a single driving noise source whose intrinsic nature is controlled by the correlation parameter c , which directly tunes the degree of correlation in the pairwise interactions, from anti-correlated ($c = -1$) to completely uncorrelated ($c = 0$). This is in contrast to systems having a center-of-mass conserving noise in addition to a thermal bath, where two independent noise sources act on the system [20]. Adding a thermal noise to random-organizing systems, however, increases the infinite-wavelength density fluctuations in an identical manner for all noise

correlations c . We have summarized this in the “Discussion” section of the main text -

Random-organizing systems are inherently athermal (Eqs. 1, 2, and 3). We investigate how their long-range structure changes at a finite temperature by incorporating thermal noise, $\sqrt{2D^{\text{th}}}\xi_i^{\text{th}}(t)$, into the generalized model (Eq. 5), where $\xi_i^{\text{th}}(t)$ is a standard Gaussian white noise, $D^{\text{th}} = k_B T/\gamma$, k_B is the Boltzmann constant, and T is the temperature. We assume that the thermal noise and the intrinsic athermal noise, $\xi_{ji}(t)$, are uncorrelated (Eqs. 5, and SI S79). We find, both analytically and in particle simulations, that the infinite wavelength density fluctuations, quantified by $S(k \rightarrow 0)$, increase with temperature, across all noise correlations c and systems (SI Sec. I.D, Eq. S81, Fig. S7). Notably, systems with anti-correlated noise ($c = -1$), which are hyperuniform when athermal, show $S(k \rightarrow 0) \propto T$, illustrating how hyperuniformity is affected by thermal fluctuations (SI Fig. S7 inset). This behavior, also observed in other non-equilibrium hyperuniform systems [20], is similar to how thermal fluctuations weaken hyperuniformity in equilibrium crystals, as described by the fluctuation-compressibility relationship $S(\mathbf{k} = \mathbf{0}) = \kappa \rho k_B T$, where κ is the isothermal compressibility [18, 30].

Moreover, even the case of uncorrelated noise ($c = 0$) in random-organizing systems *cannot* be mapped to a thermal-type additive noise. This is because the athermal noise in random-organizing systems is multiplicative, which cannot be simply replaced by an additive thermal noise even for $c = 0$ (see Eq. 5 in the main text). Indeed, when we choose $c = 0$ and $\Lambda_{ij,\alpha\beta} = a\delta_{\alpha\beta}$ to remove the multiplicative nature of the noise, Eq. 5 can be effectively reduced to a system of interacting Brownian particles in a thermal bath at equilibrium (see the discussion on “Steady-state distribution” in SI Sec.I. B.1). However, $\Lambda(\mathbf{x}, \mathbf{y})_{\alpha\beta} \neq a\delta_{\alpha\beta}$ and $c \neq 0$, in general, for random-organizing systems (see Table S1).

Finally, the universal active phase behavior in random-organizing systems stems from the fact that while the individual noise types and sources are different in all systems, they can nevertheless be approximated by a generalized model (see Eq. 5 of the main text, and SI. Sec. 1.A for the procedure of approximating the discrete-time dynamics by a continuous-time SDE). Hence, our results are agnostic to the specific type or form of noise in the original dynamics, as long as it can be mapped to the generalized model (Eq. 5).

2. Following the question above, in [44], it was found that $S(0) \sim T$ is only valid at the low temperature region ($T \ll 1$), which is similar to the effect of thermal noise on equilibrium crystals at low temperature, and I think Eq. 10 may also have the similar constraint, which should be valid only when c is near -1. Therefore, I would suggest the authors to clarify this.

We stress that Eq. 10 is valid for *all* noise correlations $c \in [-1, 0]$ when the system is athermal, as confirmed by the excellent agreement between our theory and particle and continuum simulations (Fig. 2). Upon the addition of thermal noise, we have shown that $S(k \rightarrow 0) \propto 1 + c + aT$, hence, $S(k \rightarrow 0) \propto T$ is valid only for $c = -1$ (Fig. R3).

3. I don't understand the different between RO and BRO as shown in Fig. 2, as my understanding is that both models are the same at the fixed c , in which the noise vectors have the same correlation c between i and j . I would suggest the authors to clarify the difference between the two models, if there are any.

RO has two sources of noise - (i) direction of kicks, and (ii) magnitude of kicks. BRO has a single source of noise - (i) magnitude of kicks. At a fixed c , the direction of the noise vector will be opposite in both systems. But the direction of the noise vector will always be along the line joining the center of particles in BRO, whereas in RO, it can be in any randomly chosen direction. We had already pointed this out in the main text in the section “Universal active phase behavior” - “There are three distinct sources of noise in random-organizing systems: magnitude of kicks, direction of kicks, and selection of particles. Notice that the origin of stochasticity in RO, BRO, and SGD is different; (i) for RO, the magnitude and direction of kicks are both noisy, while the selection of particles is deterministic, (ii) for BRO, the magnitude of kicks is noisy, while both the direction of kicks and selection of particles is deterministic, (iii) for SGD, the selection of particles is noisy, while the magnitude and direction of kicks are both deterministic (Figs. 1b-d, Eqns. 2, 3, and 4).”

[1] G. Rotskoff and E. Vanden-Eijnden, Communications on Pure and Applied Mathematics **75**, 1889 (2022).

[2] D. S. Dean, Journal of Physics A: Mathematical and General **29**, L613 (1996).

[3] E. Bertin, H. Chaté, F. Ginelli, S. Mishra, A. Peshkov, and S. Ramaswamy, New journal of physics **15**, 085032 (2013).

- [4] A. P. Solon, M. E. Cates, and J. Tailleur, *The European Physical Journal Special Topics* **224**, 1231 (2015).
- [5] A. Donev and E. Vanden-Eijnden, *The Journal of chemical physics* **140** (2014).
- [6] P.-H. Chavanis, *Physica A: Statistical Mechanics and its Applications* **390**, 1546 (2011).
- [7] H. Risken, *The fokker-planck equation* (1996).
- [8] P. Illien, arXiv preprint arXiv:2411.13467 (2024).
- [9] P. Illien and A. Carof, arXiv preprint arXiv:2501.16206 (2025).
- [10] V. Démery, O. Bénichou, and H. Jacquin, *New Journal of Physics* **16**, 053032 (2014).
- [11] P.-H. Chavanis, *Physica A: Statistical Mechanics and its Applications* **387**, 5716 (2008).
- [12] S. Wilken, A. Z. Guo, D. Levine, and P. M. Chaikin, *Physical review letters* **131**, 238202 (2023).
- [13] G. Zhang and S. Martiniani, arXiv preprint arXiv:2411.11834 (2024).
- [14] E. Tjhung and L. Berthier, *Physical review letters* **114**, 148301 (2015).
- [15] E. Tjhung and L. Berthier, *Journal of Statistical Mechanics: Theory and Experiment* **2016**, 033501 (2016).
- [16] K. J. Wiese, *Physical Review Letters* **133**, 067103 (2024).
- [17] X. Ma, J. Pausch, G. Pruessner, and M. E. Cates, arXiv preprint arXiv:2507.07793 (2025).
- [18] S. Torquato, *Physics Reports* **745**, 1 (2018).
- [19] D. Hexner and D. Levine, *Physical review letters* **118**, 020601 (2017).
- [20] Q.-L. Lei, M. P. Ciamarra, and R. Ni, *Science advances* **5**, eaau7423 (2019).
- [21] F. De Luca, X. Ma, C. Nardini, and M. E. Cates, *Journal of Physics: Condensed Matter* **36**, 405101 (2024).
- [22] H. Ikeda, *Physical Review E* **108**, 064119 (2023).
- [23] H. Ikeda, *SciPost Physics* **17**, 103 (2024).
- [24] Q.-L. Lei and R. Ni, *Proceedings of the National Academy of Sciences* **116**, 22983 (2019).
- [25] M. Huang, W. Hu, S. Yang, Q.-X. Liu, and H. Zhang, *Proceedings of the National Academy of Sciences* **118**, e2100493118 (2021).
- [26] L. Galliano, M. E. Cates, and L. Berthier, *Physical Review Letters* **131**, 047101 (2023).
- [27] R. L. Jack, I. R. Thompson, and P. Sollich, *Physical review letters* **114**, 060601 (2015).
- [28] Y. Kuroda and K. Miyazaki, *Journal of Statistical Mechanics: Theory and Experiment* **2023**, 103203 (2023).
- [29] S. Torquato, G. Zhang, and F. H. Stillinger, *Physical Review X* **5**, 021020 (2015).
- [30] J.-P. Hansen and I. R. McDonald, *Theory of simple liquids: with applications to soft matter* (Academic press, 2013).
- [31] S. Jastrzebski, Z. Kenton, D. Arpit, N. Ballas, A. Fischer, Y. Bengio, and A. Storkey, arXiv preprint arXiv:1711.04623 (2017).
- [32] N. S. Keskar, D. Mudigere, J. Nokedal, M. Smelyanskiy, and P. T. P. Tang, arXiv preprint arXiv:1609.04836 (2016).
- [33] Z. Xie, I. Sato, and M. Sugiyama, arXiv preprint arXiv:2002.03495 (2020).

Response to the reviewers' report

We report below a detailed response to the reviewers' reports. The original reviewer comments are in italic, and our response appears in normal font. When addressing the reviewers' comments, we also explicitly list the changes we made in the revised manuscript and highlight those changes in red here and in the revised manuscript.

Response to Reviewer 1

We thank the reviewer for their positive comments and for recommending publication of the work. Below we address the additional questions raised by the reviewer.

1. In section “Fluctuating hydrodynamic theory” of the main text, the authors discuss some previous works on coarse-graining stochastic dynamics with multiplicative noise, and mention ‘However, it has not yet been extended to systems with pairwise correlated noise between components’. As far as I understand, Ref.[48] actually addresses the case of Brownian particles with a multiplicative pairwise noise. It would be useful if the authors clarified to which extent their derivation actually differs from that of Ref.[48].

We agree with the reviewer that the ref.[48] (J Chem Phys 140, 234115 (2014)) studies the case of Brownian particles with a multiplicative, pairwise noise. However, the noise, \mathbf{B}_i in ref.[48] acts independently on each particle ($\langle \mathbf{B}_i \mathbf{B}_j \rangle \sim \delta_{ij}$), whereas in our work, the noise term ζ_{ij} is explicitly pairwise and correlated between components ($\langle \zeta_{ij} \zeta_{kl} \rangle \sim \delta_{ik} \delta_{jl} + c \delta_{il} \delta_{jk}$). This is the main feature of our work, and as we show, the correlated noise between particles, controlled by the noise correlation coefficient c , is essential for the emergence of long-range structure.

Following the suggestions of the reviewer, we have clarified the distinction between “pairwise multiplicative” and “pairwise multiplicative and correlated” noise, and the modified “Fluctuating hydrodynamic theory” section of the main text now reads - “**However, it has not yet been extended to systems where noise is both pairwise and correlated across components. Starting from Eq. 5, we extend Dean’s method [1] and its subsequent generalizations [48] to incorporate pairwise correlated noise, and derive the resulting fluctuating hydrodynamic equation for $\rho(\mathbf{x}, t)$ in arbitrary spatial dimension d (see SI Sec. I.B.2).**”

2. In the supplementary information, the authors demonstrate that their dynamics follows a Boltzmann steady-state measure at equilibrium, as it should. Here, equilibrium corresponds to choosing a specific noise statistics that amounts to enforcing some additive, diagonal noise correlations. In other words, the three models considered by the authors are generally out-of-equilibrium. Although ‘Random Organization’ and ‘Biased Random Organization’ are clearly inspired by nonequilibrium systems, it could be useful to comment further on why ‘Stochastic Gradient Descent’ should be regarded as a nonequilibrium dynamics.

Noise in Stochastic Gradient Descent (SGD) comes from the selection of particle pairs. Since at thermal equilibrium, there is no “selection” of particles, the SGD noise is inherently non-equilibrium in origin. A simple way to understand this is to look at the generalized model of random-organizing systems (Eq. 5), which approximates the dynamics of SGD. The noise term in Eq. 5 is multiplicative and correlated between particles—both of which are *not* the features of a stochastic system at thermal equilibrium. From an energetic perspective, energy is continuously added to and removed from the system as a consequence of batch selection in SGD.

Following the suggestions of the reviewer, we have added this explanation in the SI, and the modified SI Sec.I.A now reads - “**For an overdamped system of interacting particles at thermal equilibrium, the noise is additive, and uncorrelated across particles. In contrast, the noise in Eq. S13 is multiplicative and pairwise correlated between particles. This distinction provides a clear microscopic explanation for why RO, BRO, and SGD dynamics are intrinsically out of equilibrium.**”

3. In section ‘Stochastic Gradient Descent’ of the main text, the authors mention that SGD is meant to minimize the total energy. It would be insightful to show that the steady-state distribution of the stochastic dynamics in Eq.(5) is indeed highest at the minimum energy of the system. In other words, the authors should demonstrate that such a property holds for the specific choice of diffusion matrix in table S1.

Optimization methods such as gradient flow minimize the total energy. Stochastic Gradient Descent (SGD), however, is different from gradient flow because of (i) a finite learning rate, and (ii) batch selection [2]. Thus, SGD does not minimize the total energy, rather the partial energy, as mentioned in “Stochastic Gradient Descent” section of the main text, “SGD then corresponds to randomly selecting a subset of terms in E and updating the corresponding particle positions—either one or both at once—to minimize the partial energy.”

We now show that SGD does *not* strictly minimize the total energy. The total energy E is,

$$E = \frac{1}{2} \sum_i \sum_{j \neq i} V(\mathbf{x}_i, \mathbf{x}_j). \quad (\text{R1})$$

The variation in E can then be written as (up to second order in $\Delta \mathbf{x}_i$),

$$\begin{aligned} dE &= \frac{1}{2} \sum_i \sum_{j \neq i} [V(\mathbf{x}_i + \Delta \mathbf{x}_i, \mathbf{x}_j + \Delta \mathbf{x}_j) - V(\mathbf{x}_i, \mathbf{x}_j)] \\ &= \frac{1}{2} \sum_i \sum_{j \neq i} \left[\Delta \mathbf{x}_i^T \nabla_{\mathbf{x}_i} V_{ij} + \Delta \mathbf{x}_j^T \nabla_{\mathbf{x}_j} V_{ij} + \frac{1}{2} \Delta \mathbf{x}_i^T \nabla_{\mathbf{x}_i} \nabla_{\mathbf{x}_i} V_{ij} \Delta \mathbf{x}_i + \frac{1}{2} \Delta \mathbf{x}_j^T \nabla_{\mathbf{x}_j} \nabla_{\mathbf{x}_j} V_{ij} \Delta \mathbf{x}_j + \Delta \mathbf{x}_i^T \nabla_{\mathbf{x}_i} \nabla_{\mathbf{x}_j} V_{ij} \Delta \mathbf{x}_j \right]. \end{aligned} \quad (\text{R2})$$

We know that $\langle \zeta_{ij}(t) \zeta_{kl}^T(t') \rangle = \delta(t-t') [\delta_{ik} \delta_{jl} + c \delta_{il} \delta_{jk}] \mathbb{I}$, and $\Delta \mathbf{x}_i = \mathbf{x}_i(t + \Delta t) - \mathbf{x}_i(t)$ evolves according to the SDE $\frac{d\mathbf{x}_i}{dt} = -\frac{1}{\gamma} \sum_{j \neq i} \nabla_{\mathbf{x}_i} V(\mathbf{x}_i, \mathbf{x}_j) + \sum_{j \neq i} \sqrt{\Lambda_{ji}} \zeta_{ji}$. We can then write the following equations under the Ito interpretation,

$$\begin{aligned} \langle \Delta \mathbf{x}_i \rangle &= -\frac{1}{\gamma} \sum_{j \neq i} \nabla_{\mathbf{x}_i} V_{ij} \Delta t \\ \langle \Delta \mathbf{x}_i \Delta \mathbf{x}_j^T \rangle &= \delta_{ij} \sum_{k \neq i} \Lambda_{jk} \Delta t + c \Lambda_{ij} \Delta t, \end{aligned} \quad (\text{R3})$$

where terms of order higher than Δt are neglected. Note that $\Lambda_{ii, \alpha\beta} = 0$ (no self-interaction). Using Eqs. R2 and R3, we now get,

$$\frac{d\langle E \rangle}{dt} = -\frac{1}{\gamma} \sum_i \left| \sum_{j \neq i} \nabla_{\mathbf{x}_i} V_{ij} \right|^2 + \frac{1}{2} \sum_i \sum_{j \neq i} \text{Tr} \left[\nabla_{\mathbf{x}_i} \nabla_{\mathbf{x}_i} V_{ij} \left(\sum_{k \neq i} \Lambda_{ki} \right) \right] + \frac{c}{2} \sum_i \sum_{j \neq i} \text{Tr} [\nabla_{\mathbf{x}_i} \nabla_{\mathbf{x}_j} V_{ij} (\Lambda_{ij})]. \quad (\text{R4})$$

The first term in Eq. R4 always minimizes the total energy, whereas the second and third terms do not. Hence, the total energy is not strictly minimized. However, energy minima are a fixed point of the equation since the right hand side of the evolution equation vanishes, i.e., $\frac{d\langle E \rangle}{dt} = 0$, when $\nabla_{\mathbf{x}_i} \sum_{j \neq i} V(\mathbf{x}_i, \mathbf{x}_j) = 0$ for each particle i . In the regime of weak, perturbative noise (e.g., batch fraction $b_f \sim 1$), the diffusion matrix is negligible compared to the drift term. As a result, the system primarily follows the gradient-descent drift, and relaxes toward an energy minimum. **We include this discussion of the energy evolution in SI Section 1A.**

Response to Reviewer 3

We thank the reviewer for their comments. Below we address the questions of the reviewer point-by-point.

1. *I appreciate the efforts that the authors have spent in revising the manuscript and the explanation they have offered to the questions in my previous report. I agree with the authors that the influence of c is not exactly the same as the effect of thermal noise in random organising hyperuniform fluids in the literature, as they come with slightly different mathematical forms. But I think the essential physics in the effect of c is the same as adding an extra noise, and in this work it is a white noise but not exactly the same way as being added in the active hyperuniform fluids previously. I believe that the effect of c can be also decoupled as an extra noise term in addition to the noise conserving the center of the mass of the system. Then the finding in this work is not “universal” as claimed by the authors, as the effect should depend on the exact form of the extra noise, such as the color of the noise, the mathematical form of correlation in*

the noise, etc. Therefore, I don't think this manuscript is suitable for publications in high profile journals like Nature Communications, but rather suggest the authors to resubmit it (with revision) to more specialised journals.

We thank the reviewer for acknowledging that the influence of the noise correlation coefficient c in our work is not the same as the effect of thermal noise random organizing hyperuniform fluids in the literature. However, we respectfully disagree with the other two points raised and trust that our explanation below will clarify the merits of our work. The two criticisms that we address are that (i) the essential physics in the effect of c is the same as adding an extra noise, which in our work is a white noise, and (ii) the effect of c can be decoupled as an extra noise term in addition to the noise conserving the center of the mass of the system. Then the finding in our work is not “universal” and should depend on the exact form of the extra noise, such as the color of the noise, the mathematical form of correlation in the noise, etc. Below we answer both of these concerns raised by the reviewer.

(i) In our work, we do not have two noise sources. Instead, we have a *single* driving noise source whose intrinsic nature is controlled by the correlation coefficient c , which directly tunes the degree of correlation in the pairwise interactions, from completely uncorrelated ($c = 0$) to anti-correlated ($c = -1$). We show that the emergence of long-range structure depends specifically on the multiplicative and pairwise-correlated nature of this microscopically derived noise term. This is in contrast to systems having a center-of-mass conserving (“Laplacian”) noise in addition to a thermal bath, where two independent noise sources act on the system [3].

(ii) Previous work on random organizing hyperuniform fluids has looked at the effect of adding white noise to the center of mass conserving “Laplacian” noise [3]. Note that this captures the effect of adding an external thermal noise to the center of mass conserving noise. As mentioned above, in our system, we do *not* have two different noise sources. While one can phenomenologically construct a continuum description by adding white noise to the center of mass conserving “Laplacian” noise, such an approach fails to faithfully capture our system in two key respects. First, such a description is purely phenomenological, rather than derived by directly coarse-graining the microscopic equations of motion, as we do in our work. Second, even at a phenomenological level, this continuum model is fundamentally inconsistent, as it implies the presence of two independent noise sources. Indeed, following the phenomenological route renders these alternative frameworks non-universal and makes the choice of the “additional” noise term (its color, correlation structure, etc.) crucial. Conversely, our system is driven by a single stochastic source that we do not choose because we derive it. In other words, we do not “just add the noise”.

Finally, the universal active phase behavior in random-organizing systems arises from the observation that, although the microscopic noise sources and their specific forms vary across systems (RO, BRO, SGD), their dynamics can effectively be approximated by a generalized model (see Eq. 5 of the main text, and SI. Sec. 1.A for the procedure of approximating the discrete-time dynamics by a continuous-time SDE). Consequently, our conclusions hold irrespective of the specific type or form of the underlying noise (meaning random kicks, or biased random kicks, or selection noise), provided it can be mapped onto this generalized model (Eq. 5). Once again, this is not the result of a choice but the result of a coarse-graining procedure applied to systems with different noisy dynamics that leads to the same SDE (Eq. 5). In addition, our results do not depend on choice of short range potential, dimensionality, volume fraction (as long as in the active phase), and kick magnitude. As such, our claim that the behavior is universal is well founded.

[1] D. S. Dean, Journal of Physics A: Mathematical and General **29**, L613 (1996).

[2] I. Sadrtidinov, I. Klimov, E. Lobacheva, and D. Vetrov, arXiv preprint arXiv:2505.23489 (2025).

[3] Q.-L. Lei, M. P. Ciamarra, and R. Ni, Science advances **5**, eaau7423 (2019).

Response to the reviewers' report

We report below a detailed response to the reviewers' reports. The original reviewer comments are in italic, and our response appears in normal font. When addressing the reviewers' comments, we also explicitly list the changes we made in the revised manuscript and highlight those changes in red here and in the revised manuscript.

Response to Reviewer 3

We thank the reviewer for their comments. Below we address the questions of the reviewer point-by-point.

1. I have read the response from the authors, which I disagree with. They insisted that there is only one noise in the system controlling but c with $c=0$ being uncorrelated and $c=-1$ anti-correlated, but as I mentioned that it can always be decomposed into two noises with one anti-correlated responsible for the hyperuniformity and the other one responsible for deviation from hyperuniformity. This picture actually agrees with what they have seen in their simulation, which is hyperuniform to certain lengthscale and can be seen as a distorted hyperuniformity. Therefore, I think that the authors overrated the significance of this work, and I do not think that the phenomena they saw is universal, which should depend on the exact form of the other noise changing the hyperuniformity of the system. So I cannot recommend for publication in Nature Communications.

We respectfully disagree with the two points raised by the reviewer. As discussed in detail in our previous response, we again clarify the merits of our work. The reviewer's concerns are, (i) "...the noise can always be decomposed into two noises with one anti-correlated responsible for the hyperuniformity and the other one responsible for deviation from hyperuniformity. This picture actually agrees with what they have seen in their simulation, which is hyperuniform to certain lengthscale and can be seen as a distorted hyperuniformity", and (ii) "...I do not think that the phenomena they saw is universal, which should depend on the exact form of the other noise changing the hyperuniformity of the system". Below, we address both concerns.

(i) In our work, we do *not* have two noise sources. We have a *single* driving noise controlled by the correlation coefficient c , which continuously tunes the pairwise correlations from uncorrelated ($c = 0$) to anti-correlated ($c = -1$). The presence of only one noise source is evident at all length scales—both at the particle level (the second term in Eqs. 1, 2, 3, and 5) and at the continuum level (the third term in Eq. 6). If there were indeed two noise sources, Eqs. 1 – 6 would contain two distinct noise terms instead of one, which they clearly do not.

The reviewer's position likely arises from the observation that one could hypothetically construct a phenomenological model involving, for example, a Laplacian noise term and a standard divergence noise term, which would produce a similar form for the static structure factor $S(k)$. However, this line of reasoning is incorrect for two reasons. First, phenomenological descriptions are useful when a microscopic understanding is lacking; in our case, we *do* have a complete and explicit microscopic formulation. Using a phenomenological model is therefore unjustified and, in this case, wrong. Second, even if one accepts such a phenomenological representation, it would misleadingly suggest the existence of *two* independent noise sources, whereas our system contains only one (see Eqs. 1 – 6). Finally, a variety of phenomenological constructions could in principle produce the same $S(k)$, with no principled way to determine which, if any, corresponds to the actual microscopic dynamics. Our microscopic description circumvents all of these issues associated with phenomenological descriptions.

To make the above distinction concrete, consider the example of anti-correlated noise ($c = -1$) with an additional thermal noise term (Eq. S83). This system truly has two *independent* sources of noise, which appear separately at both the microscopic level (two distinct noise terms in Eq. S83) and the continuum level (two distinct noise terms in Eq. S84). Although the $S(k)$ for this system qualitatively resembles that of a random-organizing system with arbitrary $c \neq -1$ but *without* thermal noise (compare Figs. S7 and 2b), it would be incorrect to treat the two systems as equivalent merely because their structure factors look similar. One system has two distinct noise sources; the other has only one. Their similarity in $S(k)$ does not imply physical equivalence, since by this logic, all systems exhibiting similar $S(k)$ versus k scaling would have to be considered the same, which is obviously not true.

(ii) The universal active-phase behavior in random-organizing systems arises from the observation that, although the microscopic noise sources and their specific forms vary across systems (RO, BRO, SGD), their dynamics can be effectively approximated by a generalized model (see Eq. 5 of the main text and SI Sec. 1.A for the procedure used to approximate the discrete-time dynamics by a continuous-time SDE). Consequently, our conclusions hold irrespective of the specific type or form of the underlying noise (whether random kicks, biased random kicks, or selection noise),

provided that the system can be mapped onto this generalized model (Eq. 5). This universality is not a matter of *choice*, but a direct consequence of their equivalence upon coarse-graining, which leads to the same SDE (Eq. 5).

If the results truly depended on the “*exact form of the other noise*,” as the reviewer suggests, then the long-range structure for RO, BRO, and SGD would differ, since the microscopic origins of noise in the three cases are distinct (random kicks in RO, biased random kicks in BRO, and selection noise in SGD; see Eqs. 1 – 3). The fact that all three systems exhibit the same long-range structure is, in fact, strong evidence that the results are independent of the microscopic details of the noise, so long as their dynamics can be effectively approximated by the generalized model. This is not a hand-wavy assertion; it is precisely what we observe both analytically and in simulations. Moreover, our findings do not depend on the choice of short-range potential, dimensionality, volume fraction (as long as the system remains in the active phase), or kick magnitude. Thus, our assertion that the behavior is universal is well supported.

To conclude, we thank Reviewer 3 for taking the time to read multiple revisions of our manuscript and for articulating their concerns. We regret that our previous clarifications were not persuasive, but we hope that the detailed discussion above makes the merits of our theory and the basis for our universality claims fully explicit, and that the manuscript can now be allowed to move forward.